# Tokenised Flow Matching for Hierarchical Simulation Based Inference

**Giovanni Charles** [1,2]  **Cosmo Santoni** [1]  **Seth Flaxman** [2]  **Elizaveta Semenova** [1]

## Abstract

The cost of simulator evaluations is a key practical bottleneck for Simulation Based Inference (SBI). In hierarchical settings with shared global parameters and exchangeable site-level parameters and observations, this structure can be exploited to improve simulation efficiency. Existing hierarchical SBI approaches factorise the posterior yet still simulate across multiple sites per training sample; We instead explore likelihood factorisation (LF) to train from single-site simulations. In LF sampling we learn a per-site neural surrogate of the simulator and then assemble synthetic multi-site observations to amortise inference for the full hierarchical posterior. Building on this, we propose Tokenised Flow Matching for Posterior Estimation (TFMPE), a tokenised flow matching approach that supports function-valued observations. To enable systematic evaluation, we introduce a benchmark for hierarchical SBI. We validate LF and TFMPE on this benchmark and on realistic infectious disease and computational fluid dynamics models, finding well-calibrated posteriors while reducing computational cost.

## 1. Introduction

Simulation Based Inference (SBI) enables Bayesian parameter estimation when likelihoods are intractable but simulators are available. Rather than requiring closed-form density functions, SBI methods train neural networks on simulated parameter-observation pairs to approximate posteriors directly. This makes SBI particularly well suited for calibrating complex scientific simulators where traditional MCMC methods are impractical.

Many scientific applications naturally exhibit hierarchical structure, where observations arise from multiple related settings that share common properties. A hierarchical model captures this by partitioning parameters into global parameters, governing dynamics common across all settings, and local parameters, conditioning dynamics to specific observations. For example, fitting a disease transmission model to data from multiple countries requires global parameters for pathogenic properties like disease severity and local parameters for country-specific control policies (Flaxman et al., 2020). In hierarchical SBI, both the number of parameters and the simulation cost required for inference grow with the number of local settings, making simulation cost a key practical bottleneck.

**Notation.** In Bayesian Inference, we estimate a posterior distribution over some parameters $\theta$ given observational data $y$: $p(\theta|y) \propto p(y|\theta)p(\theta)$. In Simulation Based Inference, we assume that there is no closed form solution for $p(y|\theta)$, however we do have a simulator at our disposal which can generate $y$ given $\theta$.

We consider a two-level hierarchical model where $\theta$ is partitioned into global parameters $\theta_g \in \mathbb{R}^{d_g}$ and local parameters $\eta = (\eta_1, \ldots, \eta_{n_s})$ where each $\eta_s \in \mathbb{R}^{d_l}$ parameterises site $s \in \{1, \ldots, n_s\}$. A site corresponds to a single local setting, for example a country in epidemiological modelling or a patient in clinical applications.

The simulator generates observations $y_s \in \mathbb{R}^{d_y}$ for each site $s$. We denote the complete set of observations across all sites as $y = (y_1, \ldots, y_{n_s})$. In the common situation in which the global parameters serve as the prior for local parameters, we obtain the distribution: $p(\theta_g, \eta) = p(\theta_g) \prod_{s=1}^{n_s} p(\eta_s|\theta_g)$. The complete hierarchical posterior thus takes the form

$$p(\eta, \theta_g \mid y) \propto p(\theta_g) \prod_{s=1}^{n_s} p(\eta_s \mid \theta_g)\, p(y_s \mid \eta_s). \quad (1)$$

Note that this is the same hierarchical posterior proposed by Gelman et al. (2021).

Amortised posterior estimators are trained on a dataset of training samples $\mathcal{D} = \{(\theta^{(i)}, y^{(i)})\}_{i=1}^{N}$ to learn an approximate posterior $q_\phi(\theta|y)$. In the hierarchical setting, generating each training sample requires simulating all $n_s$ sites to obtain the complete observation vector $y = (y_1, \ldots, y_{n_s})$.

[1]Imperial College London, United Kingdom [2]University of Oxford, United Kingdom. Correspondence to: Giovanni Charles <gc1610@ic.ac.uk; giovanni.charles@cs.ox.ac.uk>.

*Proceedings of the 43rd International Conference on Machine Learning*, Seoul, South Korea. PMLR 306, 2026. Copyright 2026 by the author(s).

## 1.1. Factorised Inference

Existing factorised inference approaches factorise the posterior directly, requiring multiple simulations per training sample. When observations can be assumed i.i.d., the posterior factorises as

$$p(\theta \mid y) \propto p(\theta)^{1-n_s} \prod_{s=1}^{n_s} p(\theta \mid y_s), \qquad (2)$$

obtained by applying Bayes' rule twice (see Appendix A.1). Below we summarise the most common applications of Equation 2 to Simulation Based Inference.

**Compositional inference.** Many posterior estimation workflows exploit the factorisation in Equation 2 to train estimators on datasets of parameter-observation pairs $(\theta, y_s)$, enabling inference over arbitrarily sized observation sets (Radev et al., 2023; Boelts et al., 2025; Geffner et al., 2023).

**Summary networks.** Some compositional methods use permutation-invariant summary networks, such as Deep Sets (Zaheer et al., 2017), to aggregate variable-sized observation sets into fixed-dimensional representations for posterior estimation. These approaches typically train by uniformly sampling observation subsets up to size $n_s$, requiring on average $n_s/2$ simulations per training sample.

**Factorised score estimation.** Alternative compositional approaches decompose the posterior score over observation subsets and compose individual scores during sampling via annealed Langevin dynamics (Geffner et al., 2023). Fully-Factorised Neural Posterior Score Estimation (F-NPSE) trains on single observations, requiring one simulation per

sample, while Partially-Factorised variants (PF-NPSE) train on subsets of size $n_{\max}$, requiring $n_{\max}$ simulations per sample where $1 < n_{\max} < n_s$. Increasing $n_{\max}$ reduces approximation error from score composition at the cost of sample efficiency.

**Hierarchical SBI.** When applied to hierarchical models, recent methods have introduced separate estimators for global and local parameters to reduce the number of simulations required for posterior estimation (Arruda et al., 2025; Heinrich et al., 2024; Habermann et al., 2025; Rodrigues et al., 2021). In these approaches, a global estimator is trained from multiple local samples to approximate $p(\theta_g|y)$, and a local estimator is trained on singular local samples to approximate $p(\eta_s|\theta_g, y_s)$.

## 1.2. Tokenised SBI

Tokenised SBI embeds the structure of probabilistic models into sequences of tokens, enabling flexible handling of function-valued observations. SimFormer (Gloeckler et al., 2024) introduced this approach for posterior estimation, representing parameters and observations as tokens with identifiers, positional encodings, and conditioning information. Importantly, tokenisation enables modelling functional observations: data recorded at irregular, variable, or missing input coordinates, a useful technique for any application with a spatial or temporal aspect.

## 1.3. Flow Matching for Posterior Estimation

Flow matching trains continuous normalising flows by learning vector fields that transport samples from a base distri-

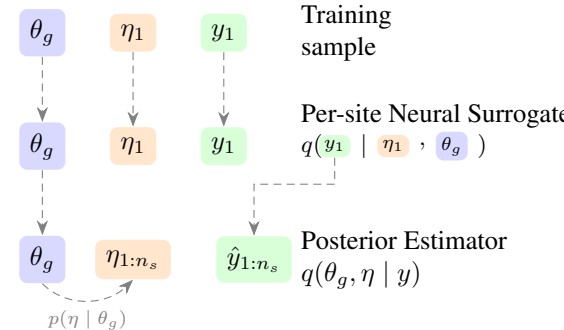

*Figure 1.* Comparison of posterior factorisation and likelihood factorisation approaches for hierarchical SBI. **Left:** Posterior-factorisation methods train from observation subsets of size at most $n_{\max}$, so each training sample requires $n_{\max}$ simulator calls (with $n_{\max} = 1$ for fully-factorised methods and $1 < n_{\max} < n_s$ for partially-factorised variants). Error in the estimated global posterior can then propagate through $\hat{\theta}_g$ to local estimation. **Right:** Likelihood factorisation trains the per-site neural surrogate $q_{\phi_l}$ from exactly one simulator call per training sample, using single-site tuples $(\theta_g, \eta_s, y_s)$, and then generates synthetic multi-site observations ($\hat{y}$) for training a separate posterior estimator $q_{\phi_p}$.

bution to a target (Lipman et al., 2023). Wildberger et al. (2023) adapted this framework to posterior estimation, learning a flow from a base distribution to $p(\theta|y)$ via the Conditional Flow Matching objective:

$$
\begin{aligned}
\mathcal{L}_{\text{FMPE}}(\phi; \theta|y) = & \\
\mathbb{E}_{t, p(\theta), p(y|\theta), p_t(\theta|\theta_1)} &||v_t(\phi, \theta_t, y) - u_t(\theta_t|\theta_1)||^2
\end{aligned}
$$

Flow matching imposes minimal architectural constraints: the learned vector field $v_t$ need only be Lipschitz continuous in $\theta$ and continuous in $t$ for uniqueness (Chen et al., 2018). Despite this flexibility, the continuity equation guarantees density estimation in the target space and bijection with the base distribution. Compared to score matching, flow matching admits simpler probability paths, such as conditional optimal transport, which leads to faster and more stable training.

**Contributions**

In this work, we combine likelihood factorisation (LF), tokenisation, and flow matching to propose a novel sample efficient method for hierarchical SBI. Our contributions are threefold:

1. We introduce LF sampling, a training strategy that factorises the likelihood, reducing the required simulations per training sample to a single site.

2. We propose TFMPE, a tokenised flow matching approach for hierarchical SBI that supports function-valued observations and benefits from the stable training dynamics of flow matching.

3. We develop the first benchmark for hierarchical SBI tasks adapted from the SBI benchmark (Lueckmann et al., 2021). This enables a systematic evaluation of the sample efficiency of TFMPE against existing hierarchical and non-hierarchical baselines. We also evaluate TFMPE on long-running, realistic hierarchical inference tasks such as a seasonal SEIR model with functional observations and a Computational Fluid Dynamics model.

## 2. Conflict of Interest

We declare that we have no competing financial interests or personal relationships that could have appeared to influence the work reported in this paper.

## 3. Method

### 3.1. Tokenisation

We adopt a tokenisation scheme that embeds arbitrarily-sized parameter and observation sets into a common latent

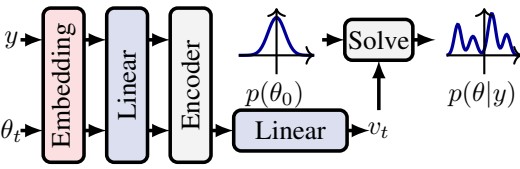

*Figure 2.* Method overview: Encoder-only transformer architecture for tokenised flow matching. This architecture template is used for both the per-site neural surrogate $q_{\phi_l}$ and posterior estimator $q_{\phi_p}$. The embedding follows the tokenisation scheme outlined in Section 3.1, while encoder performs self attention. The final linear layer outputs vector field $v_t$ optimised using the FMPE objective.

space, enabling flexible handling of variable-sized local parameter sets and function-valued observations. The complete set of model variables (global parameters $\theta_g$, local parameters $\eta$, and observations $y$) is flattened into a single sequence of $T$ tokens. For each token $i$ we record a value $v_i \in \mathbb{R}^{d_v}$ (a value of a parameter or observation, zero-padded to a fixed width $d_v$ across variables of differing dimensionality), a variable identifier $\ell_i \in \{1, \ldots, L\}$ naming the model variable the token belongs to, an intra-variable position $p_i \in \{0, \ldots, P_{\max} - 1\}$, a group identifier $g_i \in \{0, \ldots, G_{\max} - 1\}$ indicating which site the token belongs to (with a single reserved value for tokens that are not site-specific), and, when applicable, a functional input $\xi_i \in \mathbb{R}^{d_\xi}$ giving the continuous coordinate (e.g., time or spatial location) at which an observation was recorded.

**Embedding components.** Each field is mapped to a vector through a learned lookup, and the functional input is mapped through Gaussian Fourier features:

$$
\begin{aligned}
\mathbf{e}_i^{\text{id}} &= E^{\text{id}}[\ell_i] \in \mathbb{R}^{d_{\text{id}}}, \\
\mathbf{e}_i^{\text{pos}} &= E^{\text{pos}}[p_i] \in \mathbb{R}^{d_{\text{pos}}}, \\
\mathbf{e}_i^{\text{grp}} &= E^{\text{grp}}[g_i] \in \mathbb{R}^{d_{\text{grp}}}, \\
\mathbf{e}_i^{\text{fn}} &= \left[ \cos(2\pi B \xi_i); \sin(2\pi B \xi_i) \right] \in \mathbb{R}^{d_{\text{fn}}}
\end{aligned}
$$

where $E^{\text{id}}$, $E^{\text{pos}}$, $E^{\text{grp}}$ are learned embedding matrices and $B \in \mathbb{R}^{d_\xi \times d_{\text{fn}}/2}$ is a fixed random Gaussian basis that yields bounded sinusoidal features from the unbounded input $\xi_i$. A scalar flow-matching time $t \in [0, 1]$ is broadcast across tokens as $\mathbf{e}_i^{\text{t}} = t$. These are concatenated with the value $v_i$ and projected to the transformer latent dimension $d_{\text{lat}}$:

$$
\mathbf{h}_i^{(0)} = W \left[ v_i; \mathbf{e}_i^{\text{id}}; \mathbf{e}_i^{\text{pos}}; \mathbf{e}_i^{\text{grp}}; \mathbf{e}_i^{\text{fn}}; \mathbf{e}_i^{\text{t}} \right] + b, \qquad (3)
$$

with $W \in \mathbb{R}^{d_{\text{lat}} \times (d_v + d_{\text{id}} + d_{\text{pos}} + d_{\text{grp}} + d_{\text{fn}} + 1)}$. For tokens belonging to variables without functional inputs (e.g., parameters), $\mathbf{e}_i^{\text{fn}}$ is omitted from the concatenation. The variable identifier disambiguates tokens of distinct variables, and the

intra-variable position disambiguates entries within a single multi-dimensional variable. The group identifier $g_i$ distinguishes our scheme from existing tokenised SBI approaches (Gloeckler et al., 2024), which encode only variable identity and position. Making site membership an explicit input removes the need for the transformer to recover it from positional patterns alone, and is particularly helpful when local parameters and observations interact through noisy or high-variance mappings (see "No Grouping" ablation, Appendix A.5). Figure 5 in Appendix A.2 gives a worked tokenisation example together with an expanded view of the encoder-only architecture.

**Vector-field readout.** The token sequence $\mathbf{H}^{(0)} = (\mathbf{h}_1^{(0)}, \ldots, \mathbf{h}_T^{(0)})$ is processed by an encoder-only transformer with full self-attention, producing $\mathbf{H}^{(K)} \in \mathbb{R}^{T \times d_{\text{lat}}}$ after $K$ blocks. Let $\mathcal{I}_\theta \subseteq \{1, \ldots, T\}$ index the parameter tokens (those carrying entries of $\theta_g$ and $\eta$); observation tokens act purely as conditioning context. The flow-matching vector field is read out from the parameter tokens by a single shared linear projection

$$v_t(\phi, \theta_t, y)_i = W_{\text{out}} \, \mathbf{h}_i^{(K)} \in \mathbb{R}^{d_v}, \qquad i \in \mathcal{I}_\theta, \quad (4)$$

with $W_{\text{out}} \in \mathbb{R}^{d_v \times d_{\text{lat}}}$. The per-token outputs are then scattered back into the original parameter layout using the same variable identifier and position that produced the input tokens, giving a vector field with the same shape as $(\theta_g, \eta)$. The readout is shared across all parameter tokens: heterogeneity between parameters is carried by the identifier, position, and group embeddings rather than by separate output heads, which keeps the estimator amortised across different site counts and parameter layouts.

### 3.2. Likelihood Factorisation (LF) Sampling for Hierarchical Models

Rather than factorising the posterior, we propose to factorise the likelihood. By conditional independence of observations given local parameters, $p(y \mid \theta_g, \eta) = \prod_{s=1}^{n_s} p(y_s \mid \theta_g, \eta_s)$, so a per-site neural surrogate $q_{\phi_l}(y_s \mid \theta_g, \eta_s)$ of the simulator can be trained from single-site simulations. Unlike Neural Likelihood Estimation (Papamakarios et al., 2019; Hermans et al., 2020), which targets a tractable density for likelihood evaluation, here $q_{\phi_l}$ is used purely as a sampler for synthesising multi-site observations. In exchange, errors in the learned surrogate may propagate into posterior estimation. This trade-off favours LF when simulations are expensive and the per-site observation model is easier to approximate than the mapping from global parameters to full multi-site observations, whereas compositional posterior factorisation is most stable when its global estimator is trained on multiple simulations per site.

We propose LF sampling as a two-stage training strategy for amortised hierarchical posterior estimation. In

stage one, we train a per-site neural surrogate $q_{\phi_l}$ by generating training data $D_s = \{(\theta_g^{(i)}, \eta_s^{(i)}, y_s^{(i)})\}_{i=1}^N$ from single-site simulations, optimising: $\mathcal{L}_1(\phi_l) = \mathbb{E}_{\theta_g, \eta_s, y_s} [\mathcal{L}_{\text{FMPE}}(\phi_l; y_s | \theta_g, \eta_s)]$

In stage two, we train a posterior estimator $q_{\phi_p}$ on multi-site training data $D_m = \{(\theta_g^{(i)}, \eta^{(i)}, y^{(i)})\}_{i=1}^N$ where synthetic observations are sampled from the neural surrogate $q_{\phi_l}(y|\theta_g^{(i)}, \eta^{(i)})$, and optimise: $\mathcal{L}_2(\phi_p) = \mathbb{E}_{\theta_g, \eta, y} [\mathcal{L}_{\text{FMPE}}(\phi_p; \theta_g, \eta | y)]$

The surrogate and posterior estimators may use different hyperparameters or can be trained jointly in stage 2, as proposed by Gloeckler et al. (2024). We chose to train separate estimators. LF sampling can optionally be extended with sequential refinement by resampling parameters from the latest posterior to generate more concentrated training data. The complete procedure is given in Algorithm 1.

To optimise $\mathcal{L}_{\text{FMPE}}$, we adopt Gaussian probability paths $p_t(\theta|\theta_1) = \mathcal{N}(\mu_t(\theta_1), \sigma_t(\theta_1)^2 I)$ with the optimal transport vector field $u_t(\theta_t|\theta_1) = \frac{\theta_1 - (1-\sigma_{\min})\theta_t}{1-(1-\sigma_{\min})t}$.

## 4. Results

| Task | $d_g$ | $d_l$ | Total dim | Structure |
|---|---|---|---|---|
| Gaussian Linear | 1 | 5 | $1 + 5n_s$ | separated |
| Gaussian Linear Uniform | 1 | 5 | $1 + 5n_s$ | separated |
| Gaussian Mixture | 2 | 1 | $2 + n_s$ | partial pooling |
| SIR | 1 | 1 | $1 + n_s$ | separated |
| SLCP | 3 | 2 | $3 + 2n_s$ | separated |
| Two Moons | 4 | 2 | $4 + 2n_s$ | partial pooling |

*Table 1.* Hierarchical SBI benchmark tasks: global dimensionality $d_g$, per-site local dimensionality $d_l$, and total parameter dimension as a function of the number of sites $n_s$. *Separated* tasks have naturally distinct global and local parameters; *partial pooling* tasks introduce global hyperparameters governing the distribution of replicated local parameters. Full simulator formulations and priors are given in Table 6.

We evaluated TFMPE with LF sampling on a new benchmark suite of hierarchical SBI tasks as well as a more realistic infectious disease model and fluid dynamics (haemodynamics) model. Through the benchmark we investigated the scaling behaviour of TFMPE compared to non-hierarchical simulation based inference with respect to different sampling budgets and the number of local parameters. For the SEIR model in Section 4.2, we provided a more qualitative assessment of posterior estimates compared to MCMC.

### 4.1. Hierarchical SBI Benchmark

We adapted a benchmark for Simulation Based Inference (Lueckmann et al., 2021) to evaluate the sample efficiency on hierarchical inference tasks (Table 1). As in the origi-

nal benchmark, the posteriors span a range of geometries, dimensionalities, and dependence structures. Tasks are adapted under one of two hierarchical structures: *separated*, where parameters partition naturally into global and local parameters with only the local parameters replicated across $n_s$ sites; and *partial pooling* (Gelman et al., 2021), where new global parameters govern the distribution of independently replicated local parameters. Full specifications are given in Table 6. We applied automatic parameter transforms across all tasks to map constrained parameter spaces to unconstrained representations for numerical stability.

We compared TFMPE to both hierarchical and non-hierarchical baselines. The non-hierarchical baselines were Neural Posterior Estimation (NPE) (Papamakarios & Murray, 2016), Sequential Neural Posterior Estimation (SNPE) (Papamakarios et al., 2019), Simformer (Gloeckler et al., 2024), and Flow Matching Posterior Estimation (FMPE) with both MLP and transformer velocity fields (Wildberger et al., 2023); together these isolate the effects of posterior factorisation, tokenisation, sequential refinement, and flow-matching architecture. As a hierarchical baseline, we used the Posterior Factorisation (PF) method proposed by Heinrich et al. (2024), the most recent published approach with a public implementation, at the time of writing, for hierarchical Simulation Based Inference. Simformer serves here as a proxy for NPSE, providing a tokenised non-hierarchical comparator with a public implementation.

**LF** For likelihood factorisation we used our TFMPE method with an encoder-only transformer architecture, depicted in Figure 2. The per-site neural surrogate $q_{\phi_l}$ and posterior estimator $q_{\phi_p}$ shared the same architecture but were trained separately on their respective datasets and objectives. The transformer comprised 2 encoder blocks with 16 attention heads and 2 feedforward layers, operating on a latent dimension of 256. Parameters and observations were tokenised following the scheme described in Section 3.1, without functional inputs since the benchmark tasks contain only fixed-dimensional observations. No independence structure was imposed, allowing full attention between all tokens. For sampling, we solved the flow ODE using a Dormand-Prince 5(4) (Dormand et al., 1987) solver with adaptive step control and tolerances rtol = atol = $10^{-5}$. The model was trained with the Adam optimiser at learning rate $10^{-4}$, batch size 100, and early stopping with patience of 100 epochs for a maximum of 1000 epochs, following the LF sampling strategy described in Section 3.2.

**NPE** We used the NPE implementation provided by SBI v0.25.0 (Boelts et al., 2025) with its default parameterisation. As a non-hierarchical baseline, NPE was trained on the fully flattened hierarchical problem: the target variable was the concatenated parameter vector $(\theta_g, \eta_1, \ldots, \eta_{n_s})$, and

the conditioning variable was the concatenated observation vector $(y_1, \ldots, y_{n_s})$. The posterior estimator was a Neural Spline Flow (NSF) (Durkan et al., 2019) with 5 coupling transforms, 10 spline bins, and 50 hidden features per layer. Gradient descent was performed by the Adam optimiser with a learning rate of $5 \times 10^{-4}$ and early stopping with patience of 20 epochs on a 10% validation split. Both parameters and observations were independently z-scored per dimension.

**PF** For our PF baseline, we used the implementation published by Heinrich et al. (2024). The architecture comprises of two separate Masked Autoregressive Flows (MAF) for global and local parameter estimation respectively and a permutation-invariant DeepSet encoder to embed local observations for the global estimator. The DeepSet encoder used a 3-layer MLP to embed individual observations into a 128-dimensional latent space, followed by mean-max pooling and a 4-layer decoder with GELU activations. Both estimators used 6 MAF transforms, each with 2 autoregressive blocks, 256 hidden features, and tanh activation. The model was trained with AdamW (learning rate $10^{-3}$, weight decay $5 \times 10^{-5}$), with a cosine annealing schedule, batch size of 128, and early stopping on a 10% validation split with a maximum of 50 epochs. Unlike NPE, no z-score normalisation was applied to parameters or observations.

**Simformer** We used Simformer (Gloeckler et al., 2024), a score-based diffusion method that uses a transformer to estimate the score function for all-in-one posterior inference. The transformer used a token dimension of 40, condition token dimension of 10, time embedding dimension of 128, 4 attention heads, 6 layers, attention size of 10, and a widening factor of 3, with skip connections and layer normalisation enabled. Training used Adam with initial learning rate $10^{-3}$ and minimum learning rate $10^{-6}$, gradient clipping at norm 10.0, a 5% validation split, and early stopping with patience of 5 evaluation rounds. The conditioning mask function was structured random masking, and posterior samples were drawn with an Euler-Maruyama SDE solver using 500 discretisation steps.

**SNPE** We implemented Sequential Neural Posterior Estimation (Papamakarios et al., 2019) using the SBI library (Boelts et al., 2025) with the same NSF architecture as NPE. SNPE was run for 5 sequential rounds with 10 atomic proposals per round.

**FMPE (MLP)** Flow Matching Posterior Estimation (Wildberger et al., 2023) was implemented using the SBI library (Boelts et al., 2025) with an MLP-based velocity field estimator. The MLP used 5 hidden layers with 100 units each, GELU activations, layer normalisation, and skip connections. Time conditioning used a sinusoidal embed-

ding of dimension 32. Training used Adam with learning rate $5 \times 10^{-4}$, batch size 200, gradient clipping at norm 5.0, and early stopping with patience of 20 epochs on a 10% validation split.

**FMPE (Transformer)** We also evaluated FMPE with a transformer-based velocity field estimator, as provided by the SBI library. The transformer used 5 layers with 10 attention heads, latent dimension 100, and feedforward expansion ratio 4, with adaptive layer normalisation for time and context conditioning. All other training parameters were identical to FMPE (MLP).

These configurations define the comparison in Figure 3 and Tables 4 and 5.

### 4.1.1. EVALUATION

The posteriors were evaluated using $\ell$-c2st (Linhart et al., 2023), which provides a reliable metric for local posterior consistency (see Appendix A.10 for protocol details). When adapted to high-dimensional hierarchical inference tasks, standard methods for computing reference posteriors (rejection sampling and MCMC) become intractable. Many benchmark tasks contain invalid parameter regions where likelihood evaluation fails, exhibit complex multi-modal posterior geometries, or require solutions to ordinary differential equations that produce noisy gradients - posing challenges even for gradient-based MCMC samplers. $\ell$-c2st avoids these issues by using a classifier-based diagnostic that does not require explicit density evaluation or reference samples from the true posterior. Throughout this section, simulation budget $N$ counts only calls to the true simulator; samples drawn from learned surrogates such as $q_{\phi_l}(y_s \mid \theta_g, \eta_s)$ in LF sampling are treated as neural-surrogate compute

and are not charged against $N$. Numerical summaries for the budget and site-scaling sweeps are reported alongside Figure 3 in Tables 4 and 5.

Figure 3 shows a clear separation between factorised and non-hierarchical methods. Across tasks, the non-hierarchical baselines (NPE, Simformer, SNPE, and both FMPE variants) typically do not fall below $\ell$-C2ST $\geq 0.15$ as simulation budget increases, since a large number of simulator calls are consumed in multi-site sampling. In contrast, the factorised methods more efficiently benefit from additional budget: PF improves steadily on several tasks, and TFMPE with LF achieves the best sample efficiency on most tasks when the number of sites is large relative to the simulation budget. The main exception is hierarchical Two Moons, where Heinrich's PF baseline outperforms LF. The "Direct" ablation (Appendix A.5) shows that replacing surrogate samples with true simulator samples recovers calibration on Two Moons, isolating surrogate approximation error as the cause of LF's gap. The matched "PF" ablation shows that swapping factorisation alone under the TFMPE backbone does not close the gap, so Heinrich's advantage on this task is attributable to architectural or training differences rather than to the factorisation strategy itself.

The contrast between Gaussian Mixture and Two Moons illustrates this trade-off. In Gaussian Mixture, the per-site observation model is a location family centred at the local parameter, which the neural surrogate approximates accurately. Posterior factorisation faces a harder problem: the global parameters are hyperparameters governing the distribution of local parameters, so the global estimator must infer moments of a latent distribution from noisy realisations. Since observed variance conflates the true local parameter variance with observation noise, recovering the hyperpa-

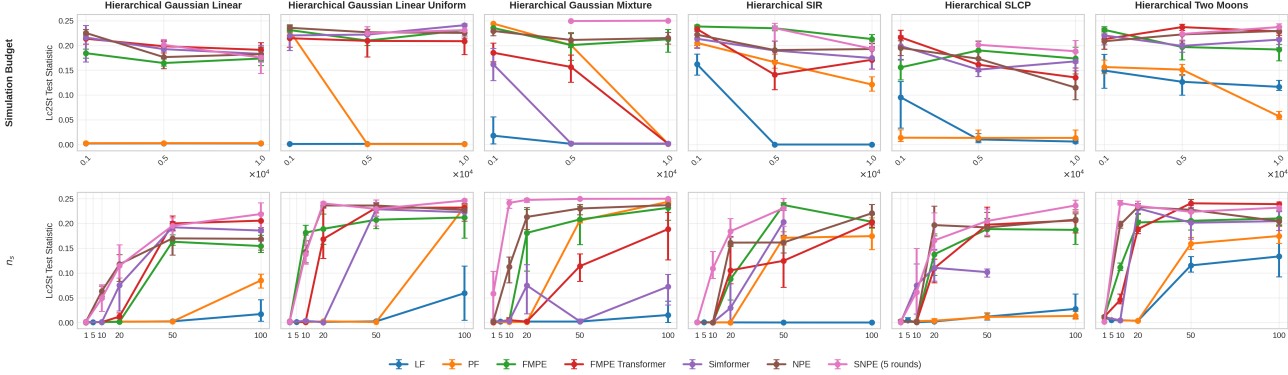

*Figure 3.* Posterior consistency measured by $\ell$-C2ST (lower is better) across the hierarchical SBI benchmark. LF: Likelihood Factorisation sampling (Section 3.2), implemented with TFMPE. PF: Posterior Factorisation as proposed by Heinrich et al. (2024). NPE: Neural Posterior Estimation (Papamakarios & Murray, 2016; Boelts et al., 2025). SNPE: Sequential Neural Posterior Estimation (Papamakarios et al., 2019). Simformer (Gloeckler et al., 2024). FMPE (MLP) and FMPE (Transformer): Flow Matching Posterior Estimation with MLP and transformer velocity fields (Wildberger et al., 2023). All results are averaged over 10 independently drawn observations per task. **Top row:** Posterior consistency as simulation budget $N$ increases, with sites fixed at $n_s = 50$ (Table 4). **Bottom row:** Posterior consistency as the number of sites $n_s$ increases, with simulation budget fixed at $N = 5{,}000$ (Table 5). Missing entries are discussed in (Table 4 and 5)

rameters requires implicit deconvolution, a problem that becomes ill-posed when observation noise dominates. Two Moons presents the opposite challenge: the nonlinear observation map produces crescent-shaped densities that are difficult to approximate, causing errors in the neural surrogate to compound during multi-site synthesis. Here PF's direct posterior estimation avoids this error propagation and achieves better calibration.

The group embedding introduced in Section 3.1 is what makes this deconvolution tractable for TFMPE at large $n_s$: it hands the attention mechanism the site partition directly rather than forcing it to be inferred. The "No Grouping" ablation (Appendix A.5) confirms that removing this signal collapses performance, with the estimator unable to recover posteriors reliably across tasks.

## 4.2. Large Scale Functional Observations in Seasonal Infectious Disease Modelling

In global infectious disease modelling, parameter estimation commonly involves irregularly observed data at many sites. Incidence data from case detection and surveillance systems varies in both spatial resolution (coordinates or administrative units) and temporal resolution (daily to weekly), with no guarantee of regularity within or between sites. Actionable prediction requires estimating disease dynamics at fine-grained resolution, yet this leads to local parameterisation at scale. Medium-sized countries alone may have hundreds of administrative units, leading to hundreds or thousands of local parameters in global studies, as defined by the Database of Global Administrative Areas (Global Administrative Areas, 2026).

We present a case study from endemic disease surveillance. The observations are simulated from a validated seasonal SEIR simulator rather than real surveillance records, so this experiment isolates inference quality from data-quality and model-misspecification effects. Incidence data are naturally function-valued: observations $y_s$ are recorded at site-specific time points $i_s$ that may vary in frequency and regularity across sites. Observation times are sampled uniformly within a time horizon $[0, T]$, and TFMPE's tokenisation scheme enables amortised inference over arbitrary observation schedules within this horizon. We estimate the global infectivity ($\beta_0$) and site-specific seasonal amplitude ($A_s$) with seasonality modelled as annually periodic (full specification in Appendix A.8). We estimate the global transmission rate $\beta_0 \sim \text{Uniform}(0.1, 2.0)$ and local seasonal amplitudes $A_s \sim \text{Uniform}(0.2, 0.5)$, with remaining parameters ($\alpha$, $\gamma$, $\phi$) held fixed.

Figure 4 compares TFMPE to MCMC-based inference for $\beta_0$ (global) and $A_s$ (local) on two sites. Visually, the marginals and pairwise structure in Figure 4 align closely with the NUTS reference posterior. Quantitatively, the two-

sample discrepancy between TFMPE and NUTS posterior draws is $\text{MMD}^2 = 0.060$ using 400 samples and 200 permutations ($p < 0.005$). Reference posteriors were computed using NUTS (Hoffman & Gelman, 2014) in TensorFlow Probability (Dillon et al., 2017), with gradients from recursive checkpointing (Kidger, 2021). Convergence required ground-truth initialisation and careful tuning (Section A.9). Beyond five sites, NUTS failed to converge reliably.

We assessed scalability by extending inference to 100 sites. At this scale, MCMC is computationally prohibitive: each proposal requires solving 100 systems of differential equations, and chains frequently fail to mix. Non-hierarchical NPE similarly requires 100 simulations per training sample. In contrast, TFMPE achieves well-calibrated posteriors from single-site simulations, as verified by the TARP calibration diagnostic (Lemos et al., 2023) (Figure 11), demonstrating that likelihood factorisation enables tractable inference at scales relevant to global surveillance.

## 4.3. TFMPE Reduces Computational Costs of Haemodynamics Calibrations

Patient-specific haemodynamics models are routinely calibrated to sparse clinical measurements (e.g. cuff pressure and MRI-derived flow waveforms) to enable prediction of unobserved quantities throughout the vasculature. Even in 1D network settings, calibration can be computationally burdensome (Blanco & Müller, 2025; Pfaller et al., 2022) because each candidate parameter set requires a full multi-cycle network solve, and the number of outlet parameters grows with network size. Recent workflows therefore rely on careful parameter subset selection and repeated forward simulations during optimisation to obtain patient-specific fits (Taylor-LaPole et al., 2025). We use this setting to illustrate how TFMPE can reduce the computational cost of calibration while preserving posterior quality. The observations are simulated from a validated 1D CFD solver calibrated to real arterial geometries rather than from real patient measurements, so this experiment isolates inference quality from clinical data quality and model-misspecification effects.

**Cost metric (simulation time vs surrogate-sampling time).** To quantify compute savings in a way that scales with the number of patients, we measure two unit costs: (i) $t_{\text{sim}}$, the wall-clock time for one CFD forward simulation for a single patient (5 cardiac cycles to allow transient startup effects to decay; final periodic cycle retained); (ii) $t_{\text{like}}$, the wall-clock time to draw one synthetic patient observation $y_s \sim q_{\phi_l}(y_s \mid \theta_g, \eta_s)$ from the per-site neural surrogate (flow-matching ODE solve). From these we report derived end-to-end costs for generating $N$ amortised training samples across $n_s$ patients: standard NPE requires $\approx N n_s\, t_{\text{sim}}$ simulator time per dataset, posterior factorisation-style subset training requires $\approx N(n_s/2)\, t_{\text{sim}}$, while TFMPE with

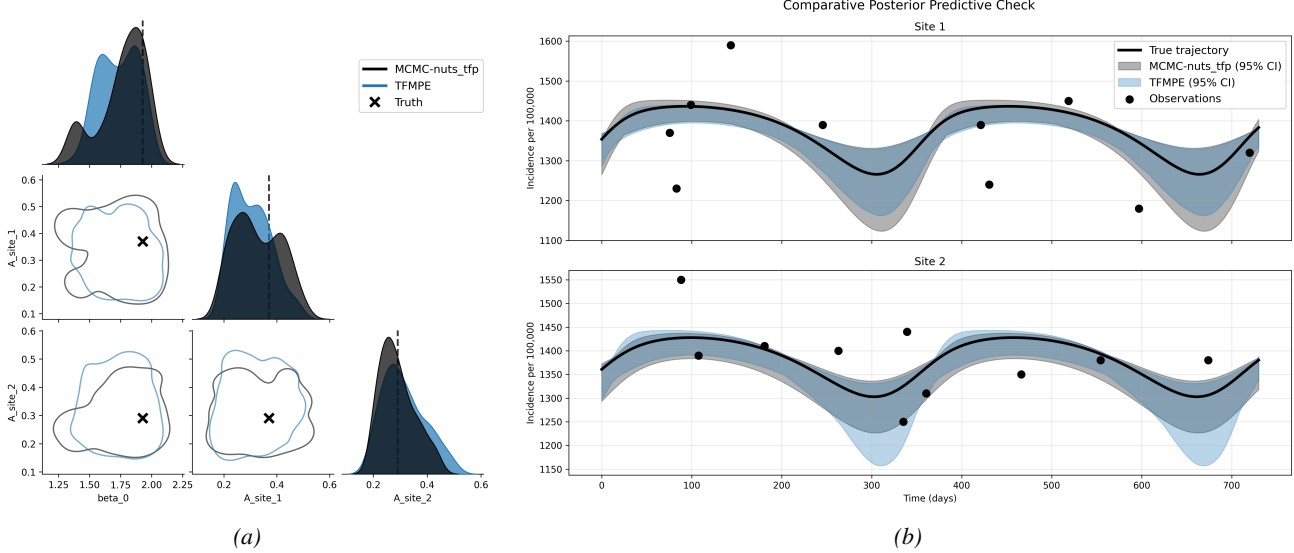

*(a)*           *(b)*

*Figure 4.* SEIR posterior comparison: a Pairplot of posterior estimates for global parameter $\beta_0$ and local parameters $A_1$, $A_2$ comparing TFMPE (tokenised flow matching) and NUTS. True parameter values shown as black crosshairs. b Posterior predictive check showing observed data (black dots) against posterior predictive samples for each method across both sites.

| **CFD forward simulator (1D blood flow network)** | |
| --- | --- |
| Unit cost | 1 patient × 5 cardiac cycles → 4 outlet flows |
| Wall time | 11.88 s    per patient (mean) |
| **TFMPE per-site neural surrogate** | |
| Unit cost | 1 draw from $q_{\phi_l}(y \mid \theta)$ via neural ODE |
| Wall time | 6.41 ms    per patient (mean) |
| **Efficiency** ($N{=}1000$ parameter draws, $n_s{=}2$ patients each) | |
| Surrogate speedup | 11.88 s / 6.41 ms = **1852×** per patient |
| NPE training cost | $N \cdot n_s \cdot 11.88\,\text{s} = 6.6\,\text{h}$ |
| TFMPE training cost | $N \cdot 11.88\,\text{s} + N \cdot n_s \cdot 6.41\,\text{ms} = 3.3\,\text{h}$ |
| End-to-end speedup | **2.0×** (grows with $n_s$) |
| **Hardware / Software** | |
| CPU / RAM | AMD EPYC 7552 (96 cores); 504 GB |
| GPU | NVIDIA A100-PCIE-40GB |
| Stack | JAX 0.8.1, Flax 0.10.6, diffrax 0.7.0 (Dopri5) |
| Precision | CFD float64; TFMPE float32 |

*Table 2.* Compute telemetry for haemodynamics experiments. TFMPE's two-stage training uses CFD only in Stage 1 ($n_s{=}1$), then replaces CFD with the learned per-site neural surrogate $q_{\phi_l}(y|\theta)$ in Stage 2, yielding 1852× faster per-patient sampling.

LF sampling requires $\approx N_1 t_{\text{sim}} + N_2 n_s\, t_{\text{like}}$ (where $N_1$ and $N_2$ are the numbers of single-site and synthesised multi-site samples, respectively). For transparency, we also report the cost of drawing posterior samples from a continuous flow. In our experiments, $t_{\text{like}} \approx 6.4$ ms compared to $t_{\text{sim}} \approx 11.9$ s, yielding a $\sim$1850× reduction in cost per synthetic patient observation.

**Compute setup and telemetry.** We report the hardware/software setup and telemetry used to compute wall-

clock and illustrative energy costs for both direct calibration and TFMPE in Table 2.

**Parameters and observations.** We use global parameters $\theta_g = (\log \beta_{\text{scale}}, \log \mu, \log Q_{\text{in}}) \in \mathbb{R}^3$ corresponding to arterial stiffness scaling, blood viscosity, and inflow amplitude. In this application, each patient constitutes a site. We use $n_s = 2$ patients for the cost telemetry in Table 2, and separately report a larger-scale calibration check with $n_s = 16$ synthetic patients in Figure 7, Table 3, and Appendix A.6. Each patient has outlet-specific Windkessel parameters (Antonuccio et al., 2021) at the four terminal vessels in the aortic-arch network, i.e. $\eta_s = (\log R_{T,s,1}, \log C_{T,s,1}, \dots, \log R_{T,s,4}, \log C_{T,s,4}) \in \mathbb{R}^8$ (implemented as Windkessel $R_2$ and $C$, with $R_1$ fixed to a small fraction of the reference resistance). Observations consist only of the four flow traces (one per terminal vessel) resampled to $N_t$ points over the final cardiac cycle, giving $y_s \in \mathbb{R}^{4N_t}$ (and $y = (y_1, \dots, y_{n_s})$).

**LF training.** Stage 1 trains the per-site neural surrogate $q_{\phi_l}(y_s \mid \theta_g, \eta_s)$ from per-patient/site tuples, where a single full-network simulation produces the four terminal-vessel flow traces for patient $s$. Stage 2 generates multi-site training observations by sampling $y \sim q_{\phi_l}(y \mid \theta_g, \eta)$ and trains the posterior estimator $q_{\phi_p}(\theta_g, \eta \mid y)$. Further simulator and prior details are provided in Appendix A.6.

**Posterior quality.** Despite the reduction in simulator calls, we verify that inferred posteriors remain well-calibrated at larger scale. In the $n_s = 16$ synthetic-patient experiment, Figure 7 shows posterior predictive checks on outlet flow

| Parameter | Site | True | Mean ± Std | 95% CI |
|---|---|---|---|---|
| **Global** | | | | |
| $\log \beta$ | — | 14.29 | $14.45 \pm 0.29$ | [13.89, 15.02] |
| $\log \mu$ | — | −5.73 | $−5.68 \pm 0.28$ | [−6.21, −5.13] |
| $\log Q_{\text{in}}$ | — | 4.64 | $4.57 \pm 0.30$ | [3.97, 5.18] |
| **Local** | | | | |
| $\log R_T$ | Desc. aorta | 19.19 | $18.89 \pm 0.38$ | [18.15, 19.63] |
| $\log C_T$ | | −20.61 | $−20.61 \pm 0.43$ | [−21.45, −19.78] |
| $\log R_T$ | Innominate | 21.53 | $21.24 \pm 0.40$ | [20.42, 21.99] |
| $\log C_T$ | | −22.96 | $−22.54 \pm 0.46$ | [−23.46, −21.65] |
| $\log R_T$ | L. carotid | 22.59 | $22.62 \pm 0.48$ | [21.54, 23.46] |
| $\log C_T$ | | −24.02 | $−23.90 \pm 0.47$ | [−24.75, −22.92] |
| $\log R_T$ | L. subclavian | 21.55 | $21.57 \pm 0.42$ | [20.70, 22.34] |
| $\log C_T$ | | −22.97 | $−22.96 \pm 0.44$ | [−23.84, −22.10] |
| 95% CI coverage | | | | **11/11** |

*Table 3.* Posterior recovery for the 16-patient haemodynamics calibration experiment. Global parameters: $\beta$ (arterial wall stiffness), $\mu$ (blood viscosity), $Q_{\text{in}}$ (inlet flow amplitude). Local parameters: Windkessel terminal resistance ($R_T$) and compliance ($C_T$) at each of the four aortic arch outlets. All 11 parameters achieve 95% credible interval coverage.

traces, and Appendix Figures 8 and 9 show the corresponding global and local posterior marginals. Samples drawn from the posterior closely track the observed waveforms, and all 11 inferred parameters achieve 95% credible interval coverage of the ground truth (Table 3), confirming that TFMPE preserves calibration quality while reducing simulator calls.

## 5. Discussion

The choice of factorisation strategy depends on the scale of local parameterisation relative to simulation budget. Standard NPE remains practical when the number of sites is small. Posterior factorisation methods offer a middle ground, achieving good sample efficiency with limited error propagation at moderate scales. LF sampling provides a distinct advantage: when local parameterisation is large and simulations are expensive, the savings from single-site training offset the approximation error in the per-site neural surrogate. The "Direct" ablation in Appendix A.5 shows that this surrogate approximation error is small on most benchmark tasks, and the matched "PF" ablation shows that swapping to LF while holding the architecture, optimiser, and schedule fixed can lead to more efficient estimation on most benchmark tasks, indicating that LF contributes sample-efficiency gains distinct from those of the tokenised flow-matching architecture itself.

The benchmark reveals a task-specific trade-off between factorisation strategies. LF sampling is preferable when the observation model is simpler to approximate than the mapping from global parameters to observations; posterior factorisation should be favoured in the converse case. Practitioners can assess this by considering whether local pa-

rameters enter the observation model through simple transformations (favouring LF) or whether global parameters induce complex, nonlinear observation structure (favouring PF). When using LF sampling, we recommend monitoring per-site surrogate loss and periodically comparing samples from the neural surrogate against held-out simulator outputs. This can be combined naturally with sequential refinement strategies such as SNLE, neural ratio estimation, or neural variational inference, and the single-site simulation budget is comparatively cheap to increase if these diagnostics reveal poor fit.

TFMPE combines three separable ideas: likelihood-factorised training, tokenised encoders for irregular hierarchical observations, and flow matching for conditional density estimation. These contributions need not be used together. The MLP ablation in Appendix A.5 shows that LF sampling remains effective without a transformer, while the "Joint" ablation shows that sharing a single estimator for both the per-site surrogate and the posterior degrades stability on several tasks.

There is nonetheless a practical scalability limit to transformer-based vector fields. Attention computation training scales quadratically with the token count. In our SEIR set-up, we experienced noticable latencies at $n_s = 100$ compared to non-transformer based alternatives. Efficient attention mechanisms can mitigate this cost, and their application to tokenised hierarchical SBI is a promising direction for future work. For the time being, in regimes with $n_s \gg 100$ we recommend an MLP-based vector field as in Arruda et al. (2025), accepting some loss in robustness in exchange for more limited compute budgets.

We released a benchmark suite to support ongoing evaluation of multiple hierarchical SBI tasks. Future work should extend the task set by adapting BernoulliGLM, Lotka-Volterra, and SLCP Distractors from the SBI benchmark (Lueckmann et al., 2021) with hierarchical structure that reflects scientifically motivated partial pooling. Introducing tasks with functional observations and local parameters would additionally enable systematic evaluation of tokenised estimators. We note that concurrent work on sample-efficient factorised posteriors (Arruda et al., 2025) and sequential refinement with truncated proposals (Deistler et al., 2022) may further improve the trade-offs explored here.

TFMPE demonstrates that tokenised flow matching scales to realistic hierarchical inference with irregular data and expensive simulators. By replacing repeated simulator evaluations with fast surrogate samples, LF sampling reduces end-to-end training cost whenever simulation dominates compute; in the haemodynamics setting we observed a $2\times$ speedup for two patients, and since flow solves are computationally negligible compared to CFD simulations.

## Impact Statement

This paper presents work towards advancing Simulation Based Inference (SBI). By proposing a new benchmark, we take steps towards reliable SBI for hierarchical models at large scales. The efficient and robust methods which we explore and evaluate on said benchmark lower the computational energy and cost of calibrating scientific simulators in domains such as epidemiology and cardiovascular medicine.

## Acknowledgements

This research was funded in whole, or in part, by the Wellcome Trust [Grant number 220900/Z/20/Z]. For the purpose of open access, the author has applied a CC BY public copyright licence to any Author Accepted Manuscript version arising from this submission. E.S. acknowledges support in part by the AI2050 program at Schmidt Sciences (Grant [G-22-64476]). We thank the Research Software for Infectious Disease Epidemiology (RESIDE) team at Imperial College London for access to their computing cluster, and SBI researchers at the Frontiers in Probabilistic Inference workshop and elsewhere for their helpful comments and feedback.

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

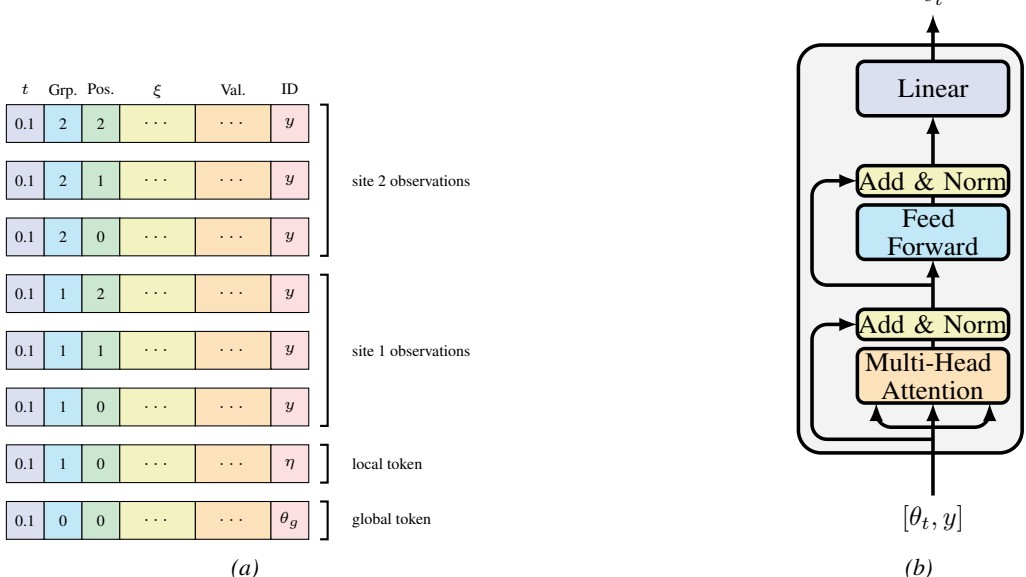

*Figure 5.* Detailed appendix figures for the method overview: a Example token layout under the Section 3.1 scheme. Each token concatenates flow time, group identifier, intra-variable position, optional functional input, value, and variable identifier before projection into the latent space. b Encoder-only architecture used by TFMPE, in which the concatenated parameter and observation tokens are processed jointly by self-attention and mapped to the vector-field output through a final linear readout.

# A. Appendix

## A.1. Derivation of the Compositional Posterior Factorisation

We derive the factorisation $p(\theta|y) \propto p(\theta)^{1-n_s} \prod_{s=1}^{n_s} p(\theta|y_s)$ for i.i.d. observations $y = (y_1, \ldots, y_{n_s})$. Applying Bayes' rule to the full posterior and using conditional independence:

$$p(\theta|y) \propto p(\theta) \prod_{s=1}^{n_s} p(y_s|\theta). \tag{5}$$

Applying Bayes' rule a second time to each single-observation likelihood:

$$p(y_s|\theta) = \frac{p(\theta|y_s)\, p(y_s)}{p(\theta)}. \tag{6}$$

We substitute (6) into (5). Since the marginals $p(y_s)$ do not depend on $\theta$, they can be absorbed into the proportionality constant:

$$p(\theta|y) \propto p(\theta) \prod_{s=1}^{n_s} \frac{p(\theta|y_s)}{p(\theta)} = p(\theta)^{1-n_s} \prod_{s=1}^{n_s} p(\theta|y_s). \tag{7}$$

Tables 4 and 5 are the appendix companion to the benchmark results in Section 4.1. Table 4 gives the exact $\ell$-C2ST means and 95% confidence intervals for the simulation-budget sweep at fixed $n_s = 50$, while Table 5 reports the corresponding site-scaling results at fixed $N = 5,000$.

## A.2. Tokenisation Example and Detailed Architecture

Figure 5 supplements Section 3.1 with a concrete token layout and an expanded view of the encoder-only architecture used by TFMPE.

## A.3. Benchmark Task Specifications

We specify the six hierarchical SBI benchmark tasks summarised in Table 1. Each task was adapted from the non-hierarchical SBI benchmark (Lueckmann et al., 2021) under one of two hierarchical structures: *separated*, where the original parameters

| Method | N = 1,000 | N = 5,000 | N = 10,000 |
|---|---|---|---|
| **Hierarchical Gaussian Linear** | | | |
| LF | **2.34e-03 ± 1.12e-03** | **2.50e-03 ± 1.16e-03** | **2.34e-03 ± 1.27e-03** |
| PF | 2.90e-03 ± 1.36e-03 | 3.00e-03 ± 1.38e-03 | 2.91e-03 ± 1.33e-03 |
| FMPE | 1.84e-01 ± 9.59e-03 | 1.65e-01 ± 7.42e-03 | 1.74e-01 ± 1.59e-02 |
| FMPE Transformer | 2.14e-01 ± 9.67e-03 | 1.98e-01 ± 1.32e-02 | 1.91e-01 ± 1.55e-02 |
| Simformer | 2.16e-01 ± 3.22e-02 | 1.93e-01 ± 1.56e-02 | 1.83e-01 ± 1.71e-02 |
| NPE | 2.25e-01 ± 8.03e-03 | 1.76e-01 ± 2.33e-02 | 1.82e-01 ± 1.52e-02 |
| SNPE (5 rounds) | – | 2.01e-01 ± 1.31e-02 | 1.76e-01 ± 2.61e-02 |
| **Hierarchical Gaussian Linear Uniform** | | | |
| LF | **1.17e-03 ± 1.09e-03** | 1.44e-03 ± 9.08e-04 | 1.39e-03 ± 8.37e-04 |
| PF | 2.29e-01 ± 3.73e-03 | **9.78e-04 ± 2.47e-04** | **1.02e-03 ± 2.64e-04** |
| FMPE | 2.31e-01 ± 1.12e-02 | 2.10e-01 ± 1.07e-02 | 2.31e-01 ± 1.03e-02 |
| FMPE Transformer | 2.15e-01 ± 1.83e-02 | 2.09e-01 ± 2.66e-02 | 2.09e-01 ± 2.53e-02 |
| Simformer | 2.20e-01 ± 2.44e-02 | 2.22e-01 ± 1.22e-02 | 2.41e-01 ± 2.59e-03 |
| NPE | 2.36e-01 ± 7.24e-03 | 2.27e-01 ± 6.36e-03 | 2.25e-01 ± 7.55e-03 |
| SNPE (5 rounds) | – | 2.25e-01 ± 1.58e-02 | 2.31e-01 ± 1.08e-02 |
| **Hierarchical Gaussian Mixture** | | | |
| LF | **1.80e-02 ± 3.18e-02** | **1.62e-03 ± 2.33e-03** | 1.57e-03 ± 2.15e-03 |
| PF | 2.44e-01 ± 1.09e-03 | 2.01e-01 ± 5.09e-03 | **1.15e-03 ± 4.63e-04** |
| FMPE | 2.35e-01 ± 8.44e-03 | 2.01e-01 ± 2.94e-02 | 2.12e-01 ± 2.43e-02 |
| FMPE Transformer | 1.85e-01 ± 2.06e-02 | 1.57e-01 ± 2.85e-02 | 1.64e-03 ± 5.97e-04 |
| Simformer | 1.62e-01 ± 3.52e-02 | 2.28e-03 ± 1.20e-03 | 2.05e-03 ± 1.21e-03 |
| NPE | 2.29e-01 ± 8.79e-03 | 2.11e-01 ± 1.61e-02 | 2.15e-01 ± 1.25e-02 |
| SNPE (5 rounds) | – | 2.49e-01 ± 1.07e-03 | 2.50e-01 ± 1.01e-04 |
| **Hierarchical SIR** | | | |
| LF | **1.62e-01 ± 2.37e-02** | **1.21e-04 ± 6.50e-05** | **1.23e-04 ± 6.09e-05** |
| PF | 2.05e-01 ± 1.12e-02 | 1.66e-01 ± 1.58e-02 | 1.21e-01 ± 1.53e-02 |
| FMPE | 2.38e-01 ± 4.36e-03 | 2.35e-01 ± 5.25e-03 | 2.13e-01 ± 9.46e-03 |
| FMPE Transformer | 2.33e-01 ± 1.03e-02 | 1.41e-01 ± 3.05e-02 | 1.71e-01 ± 2.33e-02 |
| Simformer | 2.13e-01 ± 1.82e-02 | 1.90e-01 ± 2.30e-02 | 1.75e-01 ± 2.43e-02 |
| NPE | 2.22e-01 ± 1.91e-02 | 1.91e-01 ± 2.08e-02 | 1.93e-01 ± 1.69e-02 |
| SNPE (5 rounds) | – | 2.35e-01 ± 1.63e-02 | 1.93e-01 ± 1.77e-02 |
| **Hierarchical SLCP** | | | |
| LF | 9.55e-02 ± 5.77e-02 | **1.04e-02 ± 1.17e-02** | **6.08e-03 ± 3.94e-03** |
| PF | **1.39e-02 ± 1.09e-02** | 1.35e-02 ± 1.09e-02 | 1.35e-02 ± 1.08e-02 |
| FMPE | 1.56e-01 ± 2.09e-02 | 1.90e-01 ± 2.04e-02 | 1.74e-01 ± 2.26e-02 |
| FMPE Transformer | 2.16e-01 ± 1.98e-02 | 1.61e-01 ± 1.44e-02 | 1.35e-01 ± 2.08e-02 |
| Simformer | 1.99e-01 ± 2.60e-02 | 1.51e-01 ± 1.49e-02 | 1.68e-01 ± 1.92e-02 |
| NPE | 1.95e-01 ± 1.58e-02 | 1.73e-01 ± 1.33e-02 | 1.15e-01 ± 2.94e-02 |
| SNPE (5 rounds) | – | 2.01e-01 ± 9.71e-03 | 1.88e-01 ± 2.37e-02 |
| **Hierarchical Two Moons** | | | |
| LF | **1.49e-01 ± 3.69e-02** | **1.27e-01 ± 2.68e-02** | 1.16e-01 ± 1.28e-02 |
| PF | 1.56e-01 ± 1.51e-02 | 1.51e-01 ± 1.23e-02 | **5.69e-02 ± 7.90e-03** |
| FMPE | 2.32e-01 ± 7.56e-03 | 1.97e-01 ± 2.06e-02 | 1.92e-01 ± 1.70e-02 |
| FMPE Transformer | 2.13e-01 ± 1.89e-02 | 2.37e-01 ± 6.95e-03 | 2.28e-01 ± 7.87e-03 |
| Simformer | 2.20e-01 ± 1.82e-02 | 1.99e-01 ± 1.16e-02 | 2.12e-01 ± 1.85e-02 |
| NPE | 2.08e-01 ± 1.49e-02 | 2.22e-01 ± 1.65e-02 | 2.29e-01 ± 1.15e-02 |
| SNPE (5 rounds) | – | 2.23e-01 ± 1.21e-02 | 2.37e-01 ± 9.04e-03 |

*Table 4.* $\ell$-C2ST results for simulation budget scaling ($n_s = 50$). Values show mean [95% CI] over 10 observations per task. Bold indicates best performance. SNPE (5 rounds) entries at $N = 1,000$ are omitted because the sequential schedule (5 rounds with 10 atomic proposals per round) cannot be meaningfully divided across so small a simulation budget.

| Method | $n_s = 1$ | $n_s = 10$ | $n_s = 20$ | $n_s = 50$ | $n_s = 100$ |
|---|---|---|---|---|---|
| **Hierarchical Gaussian Linear** | | | | | |
| LF | 1.13e-03 ± 6.85e-04 | 1.00e-03 ± 8.50e-04 | **1.63e-03 ± 1.05e-03** | 2.44e-03 ± 1.16e-03 | **1.71e-02 ± 2.79e-02** |
| PF | **2.43e-04 ± 4.24e-05** | 1.35e-03 ± 6.80e-04 | 1.79e-03 ± 4.94e-04 | **1.98e-03 ± 4.51e-04** | 8.49e-02 ± 1.54e-02 |
| FMPE | 4.90e-04 ± 1.69e-04 | **8.30e-04 ± 5.58e-04** | 1.64e-03 ± 6.57e-04 | 1.63e-01 ± 1.30e-02 | 1.54e-01 ± 2.07e-02 |
| FMPE Transformer | 3.43e-04 ± 2.37e-04 | 9.80e-04 ± 6.95e-04 | 1.20e-02 ± 6.96e-03 | 2.00e-01 ± 1.53e-02 | 2.06e-01 ± 1.76e-02 |
| Simformer | 5.65e-04 ± 5.03e-04 | 1.03e-03 ± 1.06e-03 | 7.52e-02 ± 5.19e-02 | 1.92e-01 ± 1.26e-02 | 1.86e-01 ± 2.26e-02 |
| NPE | 1.63e-03 ± 6.85e-04 | 6.34e-02 ± 1.24e-02 | 1.18e-01 ± 3.22e-02 | 1.70e-01 ± 3.24e-02 | 1.69e-01 ± 2.55e-02 |
| SNPE (5 rounds) | 1.04e-03 ± 5.56e-04 | 4.93e-02 ± 2.87e-02 | 1.15e-01 ± 4.01e-02 | 1.95e-01 ± 2.51e-02 | 2.19e-01 ± 3.43e-02 |
| **Hierarchical Gaussian Linear Uniform** | | | | | |
| LF | **8.82e-04 ± 2.63e-04** | 3.09e-03 ± 2.12e-03 | 1.34e-03 ± 6.12e-04 | 2.34e-03 ± 1.94e-03 | **5.92e-02 ± 6.41e-02** |
| PF | 1.36e-03 ± 4.78e-04 | **6.87e-04 ± 2.00e-04** | 2.52e-03 ± 9.82e-04 | **9.95e-04 ± 2.78e-04** | 2.33e-01 ± 6.08e-03 |
| FMPE | 1.58e-03 ± 6.85e-04 | 1.81e-01 ± 2.38e-02 | 1.89e-01 ± 3.45e-02 | 2.07e-01 ± 2.10e-02 | 2.12e-01 ± 3.97e-02 |
| FMPE Transformer | 1.74e-03 ± 1.33e-03 | 1.08e-03 ± 5.79e-04 | 1.68e-01 ± 5.89e-02 | 2.32e-01 ± 9.54e-03 | 2.32e-01 ± 1.68e-03 |
| Simformer | 1.21e-03 ± 5.36e-04 | 3.36e-03 ± 2.79e-03 | **3.65e-04 ± 2.01e-04** | 2.29e-01 ± 1.61e-02 | 2.23e-01 ± 1.55e-02 |
| NPE | 2.38e-03 ± 4.98e-04 | 1.43e-01 ± 2.56e-02 | 2.36e-01 ± 5.34e-03 | 2.36e-01 ± 5.81e-03 | 2.27e-01 ± 2.20e-02 |
| SNPE (5 rounds) | 1.74e-03 ± 1.39e-03 | 1.38e-01 ± 3.54e-02 | 2.40e-01 ± 3.82e-03 | 2.29e-01 ± 3.36e-02 | 2.46e-01 ± 2.30e-03 |
| **Hierarchical Gaussian Mixture** | | | | | |
| LF | 1.40e-03 ± 9.08e-04 | 3.86e-03 ± 4.43e-03 | 1.98e-03 ± 2.44e-03 | **2.00e-03 ± 3.07e-03** | **1.52e-02 ± 2.69e-02** |
| PF | **2.36e-04 ± 1.92e-05** | **9.65e-04 ± 2.31e-04** | **1.14e-03 ± 3.97e-04** | 2.06e-01 ± 5.08e-03 | 2.44e-01 ± 3.96e-03 |
| FMPE | 1.88e-03 ± 1.59e-03 | 3.79e-03 ± 3.57e-03 | 1.81e-01 ± 7.20e-02 | 2.08e-01 ± 4.71e-02 | 2.31e-01 ± 2.38e-02 |
| FMPE Transformer | 5.92e-04 ± 3.41e-04 | 4.81e-03 ± 4.76e-03 | 1.93e-03 ± 1.87e-03 | 1.14e-01 ± 3.01e-02 | 1.89e-01 ± 5.91e-02 |
| Simformer | 2.07e-03 ± 1.75e-03 | 3.18e-03 ± 2.66e-03 | 7.49e-02 ± 5.37e-02 | 2.37e-03 ± 3.10e-03 | 7.26e-02 ± 3.53e-02 |
| NPE | 2.24e-03 ± 5.04e-04 | 1.12e-01 ± 3.19e-02 | 2.13e-01 ± 2.40e-02 | 2.30e-01 ± 1.23e-02 | 2.37e-01 ± 5.63e-03 |
| SNPE (5 rounds) | 5.87e-02 ± 5.45e-02 | 2.42e-01 ± 1.01e-02 | 2.47e-01 ± 4.74e-03 | 2.50e-01 ± 6.83e-04 | 2.50e-01 ± 2.40e-04 |
| **Hierarchical SIR** | | | | | |
| LF | 5.82e-04 ± 4.07e-04 | 2.01e-04 ± 3.40e-05 | **3.03e-04 ± 1.56e-04** | **1.14e-04 ± 4.83e-05** | **7.33e-05 ± 1.60e-05** |
| PF | **4.29e-04 ± 2.35e-04** | 2.33e-04 ± 7.29e-05 | 3.52e-04 ± 1.18e-04 | 1.71e-01 ± 1.49e-02 | 1.74e-01 ± 3.66e-02 |
| FMPE | 5.61e-04 ± 3.81e-04 | 2.41e-04 ± 9.63e-05 | 8.76e-02 ± 6.11e-02 | 2.37e-01 ± 5.65e-03 | 2.03e-01 ± 1.60e-02 |
| FMPE Transformer | 5.38e-04 ± 3.58e-04 | 2.26e-04 ± 6.17e-05 | 1.05e-01 ± 4.70e-02 | 1.25e-01 ± 5.76e-02 | 2.03e-01 ± 1.56e-02 |
| Simformer | 5.74e-04 ± 3.84e-04 | 2.06e-04 ± 2.87e-05 | 2.96e-02 ± 5.74e-02 | 2.03e-01 ± 3.10e-02 | – |
| NPE | 5.59e-04 ± 4.03e-04 | **1.96e-04 ± 1.24e-05** | 1.61e-01 ± 1.15e-02 | 1.62e-01 ± 2.02e-02 | 2.21e-01 ± 2.23e-02 |
| SNPE (5 rounds) | 5.73e-04 ± 3.83e-04 | 1.09e-01 ± 3.23e-02 | 1.84e-01 ± 2.27e-02 | 2.31e-01 ± 3.53e-02 | – |
| **Hierarchical SLCP** | | | | | |
| LF | 1.58e-03 ± 1.59e-03 | **5.62e-04 ± 3.59e-04** | **2.12e-03 ± 2.21e-03** | 1.20e-02 ± 7.86e-03 | 2.75e-02 ± 2.77e-02 |
| PF | 6.45e-04 ± 3.52e-04 | 2.88e-03 ± 2.19e-03 | 3.77e-03 ± 2.54e-03 | **1.13e-02 ± 7.62e-03** | **1.33e-02 ± 8.86e-03** |
| FMPE | 2.43e-03 ± 2.73e-03 | 9.66e-04 ± 4.73e-04 | 1.38e-01 ± 4.88e-02 | 1.89e-01 ± 2.02e-02 | 1.87e-01 ± 3.38e-02 |
| FMPE Transformer | 2.19e-03 ± 2.59e-03 | 1.33e-03 ± 9.36e-04 | 1.10e-01 ± 2.88e-02 | 1.97e-01 ± 3.48e-02 | 2.06e-01 ± 2.36e-02 |
| Simformer | **5.61e-04 ± 4.03e-04** | 7.48e-02 ± 5.43e-02 | 1.10e-01 ± 3.46e-02 | 1.02e-01 ± 9.78e-03 | – |
| NPE | 1.73e-03 ± 1.64e-03 | 2.31e-03 ± 2.20e-03 | 1.97e-01 ± 3.60e-02 | 1.92e-01 ± 2.81e-02 | 2.08e-01 ± 2.25e-02 |
| SNPE (5 rounds) | 1.23e-03 ± 3.93e-04 | 6.16e-02 ± 8.13e-02 | 1.66e-01 ± 3.66e-02 | 2.05e-01 ± 2.91e-02 | 2.36e-01 ± 1.44e-02 |
| **Hierarchical Two Moons** | | | | | |
| LF | 1.01e-02 ± 1.01e-03 | 4.59e-03 ± 5.24e-04 | **3.56e-03 ± 1.23e-04** | **1.15e-01 ± 1.76e-02** | **1.33e-01 ± 4.80e-02** |
| PF | 1.02e-02 ± 6.04e-04 | **4.40e-03 ± 4.76e-04** | 3.67e-03 ± 6.00e-04 | 1.59e-01 ± 1.27e-02 | 1.75e-01 ± 1.92e-02 |
| FMPE | 1.10e-02 ± 8.17e-04 | 1.12e-01 ± 8.18e-03 | 2.02e-01 ± 1.93e-02 | 2.05e-01 ± 1.88e-02 | 2.10e-01 ± 1.86e-02 |
| FMPE Transformer | 1.17e-02 ± 1.15e-03 | 4.57e-02 ± 1.17e-02 | 1.89e-01 ± 1.64e-02 | 2.41e-01 ± 1.36e-02 | 2.39e-01 ± 7.48e-03 |
| Simformer | 1.09e-02 ± 8.53e-04 | 4.53e-03 ± 5.21e-04 | 2.31e-01 ± 1.05e-02 | 2.02e-01 ± 3.49e-02 | 2.05e-01 ± 2.19e-02 |
| NPE | 1.08e-02 ± 9.06e-04 | 1.97e-01 ± 7.77e-03 | 2.33e-01 ± 6.64e-03 | 2.28e-01 ± 2.49e-02 | 2.04e-01 ± 8.88e-03 |
| SNPE (5 rounds) | **1.41e-03 ± 7.44e-04** | 2.41e-01 ± 5.13e-03 | 2.35e-01 ± 1.00e-02 | 2.23e-01 ± 1.97e-02 | 2.32e-01 ± 1.10e-02 |

*Table 5.* $\ell$-C2ST results for site scaling ($N = 5{,}000$). Values show mean [95% CI] over 10 observations per task. Bold indicates best performance. Simformer entries at $n_s = 100$ on Hierarchical SIR and SLCP are omitted because its default attention implementation exceeded the 48 GB of memory available on the L40S GPUs used for these experiments at the resulting sequence lengths.

split naturally into global and site-level groups (Gaussian Linear, Gaussian Linear Uniform, SIR, SLCP); and *partial pooling*, where new global hyperparameters govern the distribution of independently replicated local parameters across sites (Gaussian Mixture, Two Moons). Table 6 lists the simulator and prior distributions for every task.

---

**Hierarchical SBI benchmark tasks**

---

**Gaussian Linear**
*Simulator*
$y_s \sim \mathcal{N}(\mu_s, \sigma^2 I)$

*Parameters*
$\sigma \sim \text{HalfNormal}(1.0), \sigma \in \mathbb{R}^+$
$\mu_s \sim \mathcal{N}(0, 1.0 I_5), \mu_s \in \mathbb{R}^5$

---

**Gaussian Linear Uniform**
*Simulator*
$y_s \sim \mathcal{N}(\mu_s, \sigma^2 I)$

*Parameters*
$\sigma \sim \text{HalfNormal}(1.0), \sigma \in \mathbb{R}^+$
$\mu_s \sim \text{Unif}(-10, 10), \mu_s \in [-10, 10]^5$

---

**Gaussian Mixture**
*Simulator*
$y_s \sim 0.5 \cdot \mathcal{N}(\eta_s, I) + 0.5 \cdot \mathcal{N}(\eta_s, 0.01 I)$

*Parameters*
$\mu_g \sim \text{Unif}(-10, 10), \mu_g \in [-10, 10]$
$\sigma_g \sim \text{HalfNormal}(1.0), \sigma_g \in \mathbb{R}_+$
$\eta_s \sim \text{TruncNorm}(\mu_g, \sigma_g, -10, 10), \eta_s \in [-10, 10]$

---

**SIR**
*Simulator*
$y_{s,t} \sim \text{Binomial}(n_c, I_{s,t}/N)$
$\frac{dS}{dt} = -\beta_s \frac{SI}{N}$
$\frac{dI}{dt} = \beta_s \frac{SI}{N} - \gamma I$
$\frac{dR}{dt} = \gamma I$

*Parameters*
$\gamma \sim \text{LogNormal}(\log 0.125, 0.2), \gamma \in \mathbb{R}^+$
$\beta_s \sim \text{LogNormal}(\log 0.4, 0.5), \beta_s \in \mathbb{R}^+$

---

**SLCP**
*Simulator*
$y_{s,j} \sim \text{MVN}([m_{0,s}, m_{1,s}], \Sigma)$
$\Sigma = \begin{bmatrix} \sigma_1^4 & \tanh(\rho)\sigma_1^2\sigma_2^2 \\ \tanh(\rho)\sigma_1^2\sigma_2^2 & \sigma_2^4 \end{bmatrix}$

*Parameters*
$\sigma_1, \sigma_2 \sim \text{Unif}(-3, 3), \sigma_1, \sigma_2 \in [-3, 3]$
$\rho \sim \text{Unif}(-3, 3), \rho \in [-3, 3]$
$m_{0,s}, m_{1,s} \sim \text{Unif}(-3, 3), m_{0,s}, m_{1,s} \in [-3, 3]$

---

**Two Moons**
*Simulator*
$y_s = f(\eta_s, \mathbf{p})$
$a \sim U(-\pi/2, \pi/2)$
$r \sim \mathcal{N}(0.1, 0.01^2)$
$\mathbf{p} = [r\cos a + 0.25, r\sin a]$

*Parameters*
$\mu_{g,0}, \mu_{g,1} \sim \text{Unif}(-1, 1), \mu_{g,0}, \mu_{g,1} \in [-1, 1]$
$\sigma_{g,0}, \sigma_{g,1} \sim \text{Unif}(0.1, 3.0), \sigma_{g,0}, \sigma_{g,1} \in [0.1, 3.0]$
$\eta_s \sim \text{TruncNorm}(\mu_g, \sigma_g, -1, 1), \eta_s \in [-1, 1]^2$

---

*Table 6.* Full simulator formulations and prior distributions for the hierarchical SBI benchmark tasks summarised in Table 1. Parameters with subscript $s$ are local parameters replicated independently for each site $s \in [n_s]$.

## A.4. LF Sampling Algorithm

---

**Algorithm 1:** LF Sampling for TFMPE

---

**Input:** Prior functions $p(\theta_g), p(\eta_s|\theta_g, h_s)$, function input distributions $p(h_s), p(i_s)$, simulator function $\text{sim}(\theta_g, \eta_s, i_s)$,
  observed data $y^{\text{obs}} = (y_1^{\text{obs}}, \ldots, y_{n_s}^{\text{obs}})$ with corresponding function inputs $h^{\text{obs}}, i^{\text{obs}}$, number of samples $N$
**Output:** Trained posterior estimator parameters $\phi_p$

// Initialization
Randomly initialize estimator parameters $\phi_l, \phi_p$ ;
Initialize empty single-simulation dataset $D_s = \{\}$ ;
Initialize empty multi-simulation dataset $D_m = \{\}$ ;

// Generate single-simulation training data
Sample $\theta_g^{(i)} \sim p(\theta_g)$ for $i = 1, \ldots, N$ ;
Sample function inputs $h_s^{(i)} \sim p(h_s), i_s^{(i)} \sim p(i_s)$ for $i = 1, \ldots, N$ ;
Sample $\eta_s^{(i)} \sim p(\eta_s|\theta_g^{(i)}, h_s^{(i)})$ for $i = 1, \ldots, N$ ;
Simulate $y_s^{(i)} \sim \text{sim}(\theta_g^{(i)}, \eta_s^{(i)}, i_s^{(i)})$ for $i = 1, \ldots, N$ ;
$D_s = \{(\theta_g^{(i)}, \eta_s^{(i)}, y_s^{(i)}, h_s^{(i)}, i_s^{(i)})\}_{i=1}^N$ ;

// Stage 1: Learn per-site neural surrogate
Optimize $\phi_l$ using $L_1(\phi_l) = L_{\text{FMPE}}(\phi_l; y_s|\theta_g, \eta_s, h_s, i_s)$ on $D_s$ ;

// Generate multi-simulation training data using the learned surrogate
**for** $i = 1, \ldots, N$ **do**
  Sample $\theta_g^{(i)} \sim p(\theta_g)$ ;
  Sample function inputs $h^{(i)} \sim p(h), i^{(i)} \sim p(i)$ ;
  Sample $\eta_s^{(i)} \sim p(\eta_s|\theta_g^{(i)}, h_s^{(i)})$ for $s = 1, \ldots, n_s$ ;
  Generate $y_s^{(i)} \sim q_{\phi_l}(y_s|\theta_g^{(i)}, \eta_s^{(i)}, i_s^{(i)})$ for $s = 1, \ldots, n_s$ ;
**end**
$D_m = \{(\theta_g^{(i)}, \eta^{(i)}, y^{(i)}, h^{(i)}, i^{(i)})\}_{i=1}^N$ ;

// Stage 2: Learn full posterior
Optimize $\phi_p$ using $L_2(\phi_p) = L_{\text{FMPE}}(\phi_p; \theta_g, \eta|y, h, i)$ on $D_m$ ;
**return** $\phi_p$ *such that* $\eta, \theta_g \sim q_{\phi_p}(\theta_g, \eta|y^{obs}, h^{obs}, i^{obs})$

---

## A.5. Ablation Experiments

TFMPE introduces many novel components to SBI, we present the relative effects of these components in Figure 6. The experimental set up for each ablation is detailed below:

**Direct**  The posterior estimator was trained on $n$ simulations per data point from the simulator instead of from the per-site neural surrogate. This experiment shows a rough estimate of the effects of surrogate approximation error on TFMPE's posterior estimates.

**Joint**  The per-site surrogate and posterior estimator were trained using the same architecture. The data set for posterior estimation is the concatenation of the surrogate training data and the posterior training data sampled from the estimator itself. This joint inference variant provides a unified estimator for sampling observations or posterior samples.

**MLP**  The transformer was replaced with a multi-layer perceptron (MLP). The MLP inputs were computed as a concatenation of the value and functional input of each token into a single vector (positional, group and variable id embeddings were omitted as redundant). The MLP consisted of $2 \times 256$ layers with rectified linear units as activations and a 10% drop out per layer. This experiment shows the relative effects of the Transformer architecture on inference.

**No Grouping**  This ablation removes the group id embedding from the input tokens. The group embedding separates

tokens belonging to each site without requiring the architecture to deduce it from the multi-site data and the positional embeddings alone.

**PF** This ablation replaces LF sampling with posterior factorisation (Equation 2) while keeping the TFMPE tokenised flow-matching backbone, group embeddings, optimiser, and schedule fixed. Two TFMPE estimators are trained sequentially: a global estimator $q_{\phi_g}(\theta_g \mid y)$ on simulations with variable site counts $n \in \{1, \ldots, n_s\}$ (drawn via stick-breaking so the simulation budget is spent exactly), and a local estimator $q_{\phi_l}(\eta_s \mid \theta_g, y_s)$ on single-site slices of the same simulations conditioned on samples from the trained global estimator. This isolates the effect of swapping LF for PF from the architectural and training differences between TFMPE and the PF baseline of Heinrich et al. (2024).

The ablation results broadly show that TFMPE performs best over the hierarchical SBI benchmark with a few caveats. The MLP ablation showed close agreement with TFMPE, making it a strong candidate for a computationally efficient alternative. However, in Gaussian Linear Uniform and SLCP it exhibited less stable posterior consistency and in the Gaussian Mixture task it reliably failed to capture the posterior. This is likely due to the fixed inferential bottleneck inherent to the MLP not being able to capture the site specific parameters. If using the MLP, we recommend tuning the model size to each particular problem. The "Direct" experiment revealed that little inconsistency is due to surrogate approximation error, as its metrics closely track TFMPE's. The "Joint" experiment uncovered delayed convergence for Gaussian Linear tasks and unstable estimators for SLCP, when the estimators were combined into a unified estimator. The "No Grouping" ablation removes the group id embedding, forcing the model to recover site membership from positional patterns alone; this collapses performance broadly across the benchmark, confirming that the group embedding is a load-bearing component of the tokenisation scheme. The "PF" ablation isolates the effect of the factorisation strategy: because it reuses the TFMPE backbone and only swaps LF sampling for posterior factorisation, any gap with TFMPE is attributable to the factorisation itself rather than to architectural or optimiser differences. It degrades performance reliably across the benchmark relative to LF sampling, which establishes LF as the driver of TFMPE's sample-efficiency advantage on these tasks. Residual differences between the PF ablation and the Heinrich et al. (2024) PF baseline in Figure 3 go in both directions across tasks and reflect architectural and training choices (transformer vs. MAF+DeepSet, flow matching vs. normalising flows, optimiser and schedule); disentangling these is a promising direction for future work.

### A.6. 1D Hemodynamics CFD Simulator

We include a networked 1D arterial hemodynamics example as a realistic hierarchical SBI task. The simulator returns function-valued outputs (flow and pressure traces) and has many outlet-specific parameters, making it a natural setting for tokenised amortised inference. The observations are synthetic data generated by a validated 1D CFD solver calibrated to real arterial geometries, rather than real patient measurements.

**Governing equations and vessel model.** We use a standard 1D pulse-wave model for area $A(x, t)$ and flow $Q(x, t)$ on each vessel:

$$\frac{\partial A}{\partial t} + \frac{\partial Q}{\partial x} = 0, \tag{8}$$

$$\frac{\partial Q}{\partial t} + \frac{\partial}{\partial x}\left(\alpha \frac{Q^2}{A}\right) + \frac{A}{\rho}\frac{\partial p}{\partial x} = -\frac{K_R Q}{A}, \tag{9}$$

with momentum-flux coefficient $\alpha = 1$ in our solver. Pressure is related to area by

$$p(A) - p_{\text{ext}} = \frac{\beta}{A_0}\left(\sqrt{A} - \sqrt{A_0}\right), \tag{10}$$

where $A_0$ is the reference area and $\beta$ is a vessel stiffness parameter. Terminal outlets use RCR (Windkessel) boundary conditions.

**Hierarchical parameters.** Global parameters are $\theta_g = (\log \beta_{\text{scale}}, \log \mu, \log Q_{\text{in}})$, where $\beta_{\text{scale}}$ rescales a baseline stiffness profile, $\mu$ is blood viscosity, and $Q_{\text{in}}$ sets inflow amplitude. We treat each *patient* as a site $s \in \{1, \ldots, n_s\}$. Each patient has terminal outlet parameters at the four terminal vessels, i.e. $\eta_s = (\log R_{T,s,1}, \log C_{T,s,1}, \ldots, \log R_{T,s,4}, \log C_{T,s,4}) \in \mathbb{R}^8$ (implemented as Windkessel $R_2$ and $C$, with $R_1$ fixed).

**Priors.** We use LogNormal priors on global parameters: $\log \beta_{\text{scale}} \sim \mathcal{N}(\log \beta_{\text{mean}}, 0.3^2)$, $\log \mu \sim \mathcal{N}(\log 0.004, 0.2^2)$, $\log Q_{\text{in}} \sim \mathcal{N}(\log 85, 0.2^2)$. Local priors are centred around structured-tree-inspired reference values $(R_{T,\text{ref}}(s), C_{T,\text{ref}}(s))$ with anti-correlated perturbations to keep the outlet time constant $\tau_s = R_{T,s}C_{T,s}$ approximately stable:

$$\log R_{T,s,o} = \log R_{T,\text{ref}}(o) + 0.4\epsilon_{s,o}, \qquad \log C_{T,s,o} = \log C_{T,\text{ref}}(o) - 0.4\epsilon_{s,o}, \qquad \epsilon_{s,o} \sim \mathcal{N}(0,1),$$

for patient $s$ and outlet index $o \in \{1, \ldots, 4\}$.

**Observations and noise model.** We condition only on the four flow traces (one per terminal vessel), each after simulating 5 cardiac cycles and resampling the final cycle to $N_t$ points, giving $y_s \in \mathbb{R}^{4N_t}$ for patient $s$. We use heteroscedastic Gaussian noise on each flow trace with standard deviation $\sigma_{s,o} = \varepsilon \max_t |Q_{s,o}(t)|$ (with $\varepsilon = 0.05$).

**LF sampling in the CFD setting.** Stage 1 trains the per-site neural surrogate $q_{\phi_l}(y_s \mid \theta_g, \eta_s)$ using per-patient tuples, where a full-network run for patient $s$ yields the four outlet flow traces comprising $y_s$. Stage 2 generates multi-site training observations by sampling $y \sim q_{\phi_l}(y \mid \theta_g, \eta)$ and trains the posterior estimator $q_{\phi_p}(\theta_g, \eta \mid y)$ using a tokenised estimator as described in Section 3.1.

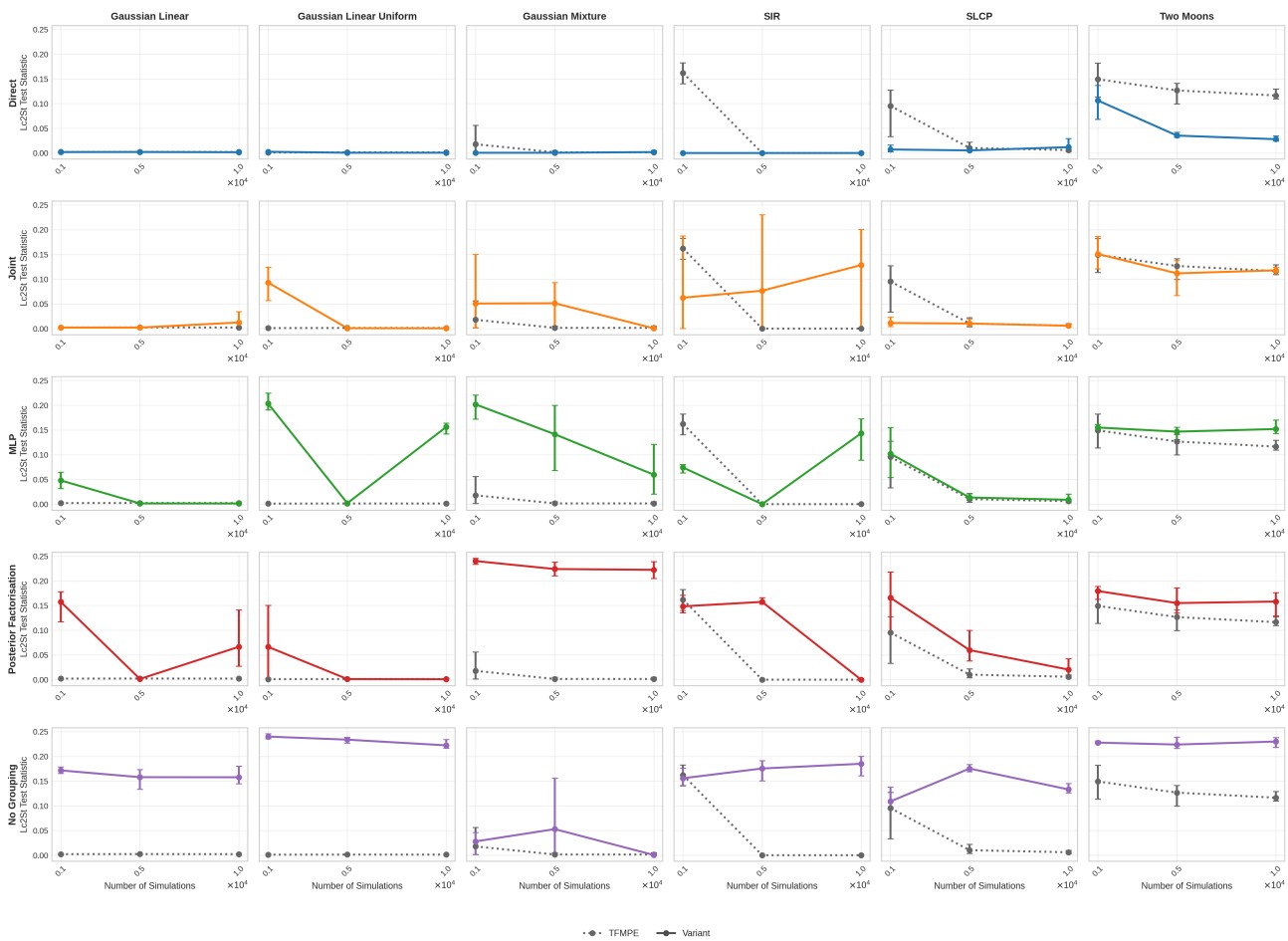

*Figure 6.* Posterior consistency measured by $\ell$-C2ST (lower is better) across the hierarchical SBI benchmark for TFMPE ablations. Each experiment is described in Section A.5. All results are averaged over 10 independently drawn observations per task. Posterior consistency was measured as the simulation budget $N$ increased, with sites fixed at $n_s = 50$.

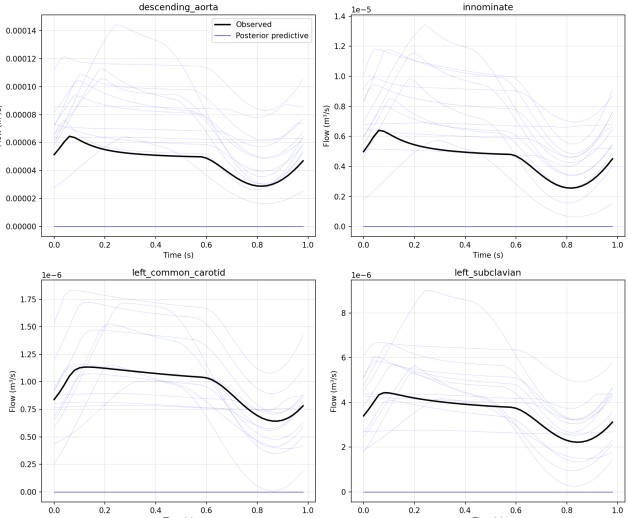

*Figure 7.* Posterior predictive check for haemodynamics calibration showing observed outlet flow waveforms (black) against posterior predictive samples (blue) across the four terminal vessels of the aortic-arch network: **A** descending aorta, **B** innominate artery, **C** left common carotid artery, and **D** left subclavian artery. All 11 inferred global and local parameters achieve 95% CI coverage (Table 3).

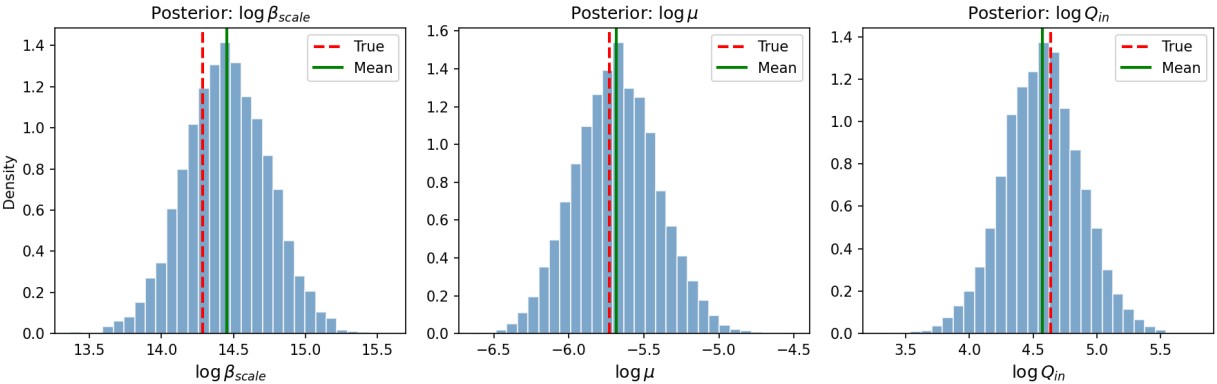

*Figure 8.* Global-parameter posterior for the 16-patient haemodynamics calibration experiment. TFMPE posterior samples align with the simulator ground truth for $\log \beta_{\text{scale}}$, $\log \mu$, and $\log Q_{\text{in}}$.

## A.7. Scalability to Larger Patient Cohorts

**Cohort expansion and computational scaling.** To evaluate scalability, the haemodynamics calibration task is extended to a cohort of $n_s = 16$ patients. Because Stage 1 likelihood training is performed on isolated patient sites (Section 4.3), the computational cost of multi-cycle CFD simulator calls is amortised. This amortisation significantly reduces the overall computational burden, making inference on larger patient cohorts tractable.

**Posterior recovery at scale.** Posterior quality is preserved at $n_s = 16$ without requiring proportional increases in the simulator budget. Posterior predictive samples closely track the observed multi-site waveforms for the expanded cohort (Figure 10). All 11 inferred global and local parameters maintain 95% credible interval coverage of the ground truth (Table 7).

| Parameter | Site | True | Mean ± Std | 95% CI |
|---|---|---|---|---|
| **Global** | | | | |
| $\log \beta$ | — | 14.29 | $14.45 \pm 0.29$ | [13.89, 15.02] |
| $\log \mu$ | — | $-5.73$ | $-5.68 \pm 0.28$ | $[-6.21, -5.13]$ |
| $\log Q_{\mathrm{in}}$ | — | 4.64 | $4.57 \pm 0.30$ | [3.97, 5.18] |
| **Local** | | | | |
| $\log R_T$ | Desc. aorta | 19.19 | $18.89 \pm 0.38$ | [18.15, 19.63] |
| $\log C_T$ | | $-20.61$ | $-20.61 \pm 0.43$ | $[-21.45, -19.78]$ |
| $\log R_T$ | Innominate | 21.53 | $21.24 \pm 0.40$ | [20.42, 21.99] |
| $\log C_T$ | | $-22.96$ | $-22.54 \pm 0.46$ | $[-23.46, -21.65]$ |
| $\log R_T$ | L. carotid | 22.59 | $22.62 \pm 0.48$ | [21.54, 23.46] |
| $\log C_T$ | | $-24.02$ | $-23.90 \pm 0.47$ | $[-24.75, -22.92]$ |
| $\log R_T$ | L. subclavian | 21.55 | $21.58 \pm 0.42$ | [20.70, 22.34] |
| $\log C_T$ | | $-22.97$ | $-22.96 \pm 0.44$ | $[-23.84, -22.10]$ |
| 95% CI coverage | | | | **11/11** |

*Table 7.* Posterior recovery for haemodynamics calibration scaled to $n_s = 16$ patients. All parameters successfully achieve 95% credible interval coverage.

## A.8. SEIR Model Formulation

The seasonal SEIR model uses a standard compartmental structure with Poisson-distributed observations:

$$y_s(t) \sim \mathrm{Poisson}(\alpha E_s(t)), \quad t \in i_s$$

$$\frac{dS_s}{dt} = -\lambda_s S_s$$

$$\frac{dE_s}{dt} = \lambda_s S_s - \alpha E_s$$

$$\frac{dI_s}{dt} = \alpha E_s - \gamma I_s$$

$$\frac{dR_s}{dt} = \gamma I_s$$

$$\lambda_s = \frac{\beta_s I_s}{N_s}$$

$$\beta_s = \beta_0 \left(1 + A_s \sin\left(\frac{2\pi t}{365} - \phi\right)\right)$$

where $\lambda_s$ is the force of infection and $y_s(t)$ represents daily incidence at site $s$ observed at times $i_s \sim \mathrm{Uniform}(0, T)$. Here $\alpha$ is the latent-to-infectious progression rate, so $\alpha^{-1}$ is the mean incubation period; $\gamma$ is the recovery rate, so $\gamma^{-1}$ is the mean infectious period; and $\phi$ is the shared seasonal phase offset controlling when transmission peaks within the year. In Section 3.2 these quantities are held fixed, while inference targets the global baseline transmission $\beta_0$ and the site-specific seasonal amplitudes $A_s$.

## A.9. SEIR Model Inference with MCMC

**Hyperparameters.** Table 8 summarises the NUTS configuration used for reference posterior computation.

**Convergence diagnostics.** Table 9 reports convergence diagnostics for the SEIR reference posteriors. The elevated $\hat{R}$ values (particularly for $\beta_0$) and low effective sample sizes indicate challenging posterior geometry, motivating the use of ground-truth initialisation.

| Parameter | Value |
|---|---|
| Sampler | NUTS (TensorFlow Probability) |
| Number of chains | 4 |
| Warmup/burn-in | 100 |
| Post-warmup samples | 100 |
| Step size | 0.1 |
| Initialisation | Ground-truth |
| Target parameters | $\beta_0, A$ |

*Table 8.* NUTS hyperparameters for SEIR reference posteriors.

| Parameter | Mean | SD | HDI 3% | HDI 97% | ESS (bulk) | $\hat{R}$ |
|---|---|---|---|---|---|---|
| $A_1$ | 0.328 | 0.086 | 0.206 | 0.480 | 39 | 1.10 |
| $A_2$ | 0.287 | 0.062 | 0.200 | 0.403 | 77 | 1.04 |
| $\beta_0$ | 1.804 | 0.162 | 1.411 | 1.998 | 9 | 1.46 |

*Table 9.* MCMC convergence diagnostics for SEIR model.

## A.10. $\ell$-C2ST Evaluation Protocol

We evaluate posterior quality using the local classifier two-sample test ($\ell$-C2ST) (Linhart et al., 2023), which diagnoses whether samples from the approximate posterior $q_\phi(\theta \mid x)$ are distinguishable from samples drawn from the joint $p(\theta, x)$ conditioned on the same observation.

**Classifier architecture.** We use fully-connected neural networks with two hidden layers of 32 units and ReLU activations. Each classifier $d_\psi : \mathbb{R}^{d_x + d_\theta} \to [0, 1]$ is trained using the Adam optimiser with learning rate $3 \times 10^{-4}$, a batch size of 100, and early stopping with patience of 100 epochs and a minimum improvement threshold of $10^{-2}$ on a 90/10 train/validation split, for a maximum of 1000 epochs.

**Test statistic.** The test classifier is trained as an ensemble of 10 classifiers over 10 cross-validation folds to distinguish concatenated pairs $(x, \theta)$ from the joint distribution (class 0) versus $(x, \theta_q)$ where $\theta_q \sim q_\phi(\theta \mid x)$ (class 1). Given a held-out observation $x_{\mathrm{obs}}$ and posterior samples $\{\theta_i\}_{i=1}^n \sim q_\phi(\theta \mid x_{\mathrm{obs}})$, the test statistic is the mean squared deviation from chance:

$$\hat{t}_{\mathrm{MSE}} = \frac{1}{n} \sum_{i=1}^n \left( d_\psi(x_{\mathrm{obs}}, \theta_i) - \frac{1}{2} \right)^2. \tag{11}$$

**Null distribution.** The null distribution is constructed from an ensemble of $H = 100$ classifiers $\{d_\psi^{(h)}\}_{h=1}^H$, each trained on independently permuted class labels. This yields null statistics $\{\hat{t}_{\mathrm{null}}^{(h)}\}_{h=1}^H$ computed identically to the test statistic.

$p$-**value.** The $p$-value is the proportion of null statistics exceeding the test statistic:

$$\hat{p} = \frac{1}{H} \sum_{h=1}^H \mathbf{1}\left( \hat{t}_{\mathrm{null}}^{(h)} \geq \hat{t}_{\mathrm{MSE}} \right). \tag{12}$$

## A.11. Calibration of Large Scale SEIR Parameter Inference

We ran the Test of Accuracy with Random Points (TARP) diagnostic, proposed by Lemos et al. (2023), to evaluate the calibration of TFMPE posterior estimates for the SEIR task with 100 sites. The results are summarised in Figure 11.

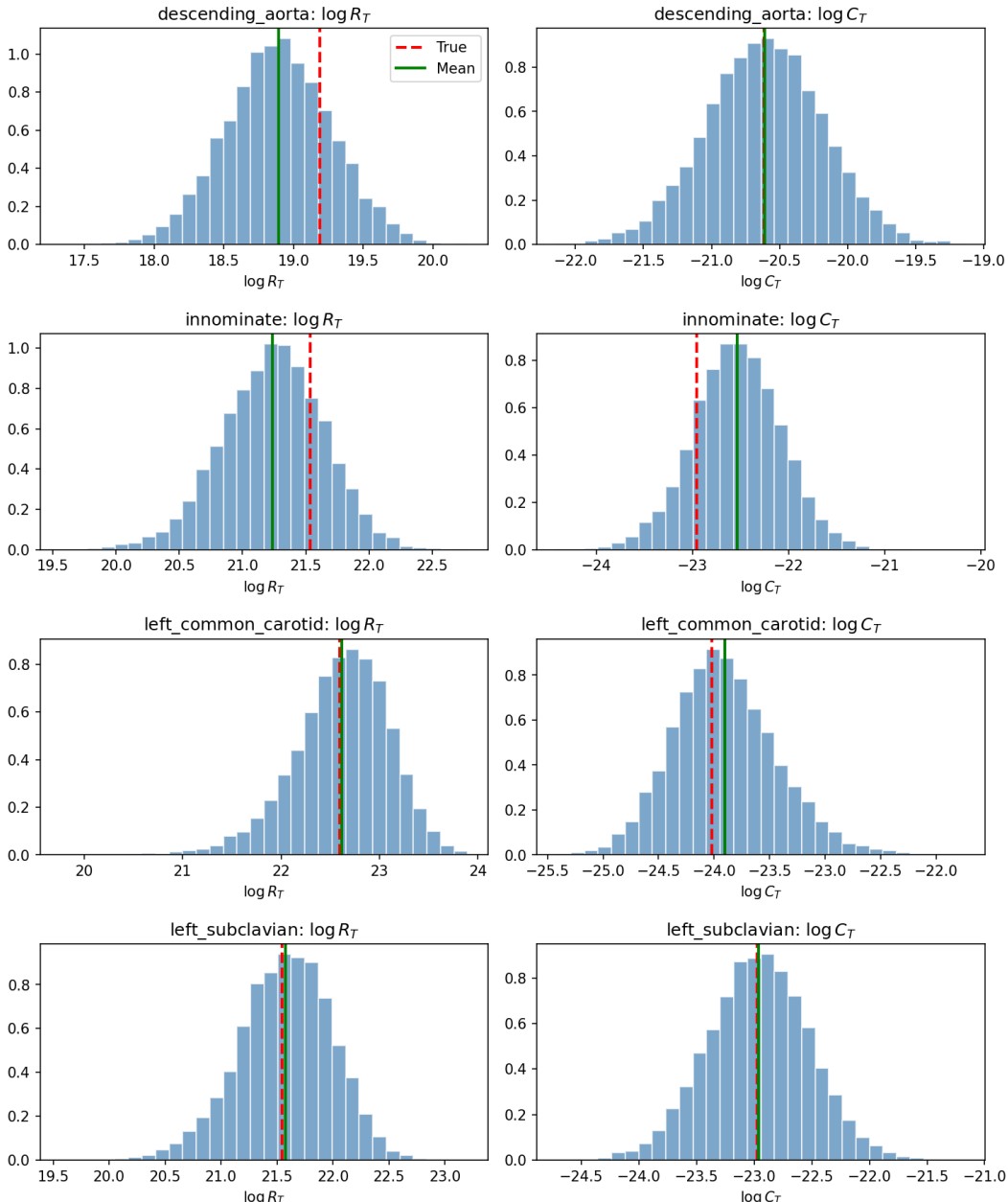

*Figure 9.* Outlet-specific local-parameter posterior for the 16-patient haemodynamics calibration experiment. TFMPE posterior samples remain concentrated around the simulator ground truth across all Windkessel resistance and compliance parameters.

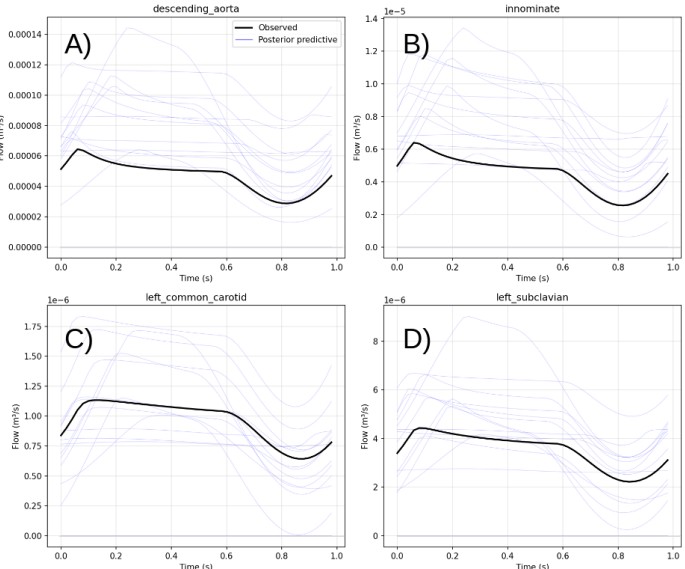

*Figure 10.* Posterior predictive check for the scaled $n_s = 16$ haemodynamics cohort across the four terminal vessels of the aortic-arch network: **A** descending aorta, **B** innominate artery, **C** left common carotid artery, and **D** left subclavian artery. Observed outlet flow waveforms (black) are shown against posterior predictive samples (blue). All 11 inferred global and local parameters achieve 95% CI coverage.

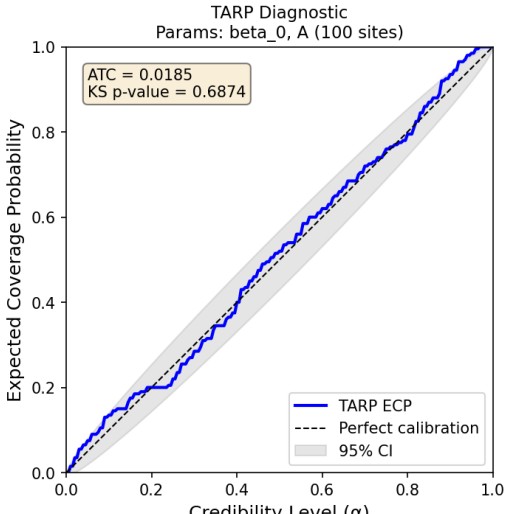

*Figure 11.* TARP calibration diagnostic for TFMPE on the seasonal SEIR model with 100 sites. TARP metrics: ATC = 0.0382, KS p-value = 0.0097.

