# OpenReview forum: "Tokenised Flow Matching for Hierarchical Simulation Based Inference"
_ICML.cc/2026/Conference — ICML 2026 regular_

### Official Review · Reviewer_3tvg · 2026-02-21

**Soundness:** 3
**Presentation:** 3
**Significance:** 4
**Originality:** 3
**Overall Recommendation:** 5
**Confidence:** 4

**Summary:**

The paper targets a practical bottleneck in simulation-based inference (SBI): the high cost of simulator calls, especially in hierarchical models with shared global parameters and many exchangeable site-level latent parameters and observations. The key idea is to exploit the natural likelihood factorization over sites to reduce simulation cost. Instead of simulating multi-site datasets during training, the method learns a per-site neural likelihood, then assembles synthetic multi-site observations by sampling from this learned likelihood. These synthetic multi-site samples are used to amortize training of a hierarchical posterior estimator without additional simulator evaluations. On top of this pipeline, the paper proposes a tokenised flow-matching model intended to handle variable numbers of sites via a tokenised representation. The authors also introduce a benchmark suite for hierarchical SBI and evaluate on the benchmark plus two application settings, reporting calibrated posteriors with reduced simulation budgets.

**Compliance With Llm Reviewing Policy:**

Affirmed.

**Key Questions For Authors:**

1. What exactly is the NPE trained on in the hierarchical setting? Only the local variables?
2. Budget accounting: when reporting “simulation budget,” do you count only calls to the true simulator, or also samples drawn from the learned neural likelihood? How many neural-likelihood samples are used to train the hierarchical posterior estimator, and how sensitive are results to that number? Does the posterior estimator scale beyond 100 sites?
3. Is the improvement over PF coming from the architecture of the estimator or the training scheme? Please include a controlled comparison between direct hierarchical Tokenised Flow Matching trained on multi-site simulations and your LF pipeline, ideally holding architectures and training compute as comparable. This would help to get a better understanding of the approximation error due to first learning a neural likelihood.
4. The authors argue that „benchmark tasks contain invalid parameter regions where likelihood evaluation fails, exhibit complex multi-modal posterior geometries, or require solutions to ordinary differential equations that produce noisy gradients“. Why is this then a good benchmark? Since your proposed benchmark are rather low dimensional (besides the number of sites) at least for some and a low number of sites MCMC should be possible to generate „ground truth“ posterior samples similar to the benchmark by Lueckman et al.
5. Regarding example in section 3.2 and 3.3: is this real data here, if not is it possible to include some real data experiments? Figure 4 is nowhere referenced, how many sites are used there to compute global estimates, 2? I miss here a qunatitative comparison for TFMPE vs MCMC. This should be extended.

**Limitations:**

Yes, limitations are mostly discussed. As far as I understood, there is no application to real data yet and maybe scalability to more sites than 100 could be discussed a bit more.

**Strengths And Weaknesses:**

Soundness
The likelihood-factorization perspective is technically well-motivated for hierarchical models where observations are conditionally independent given local and global parameters. Learning a per-site likelihood from single-site simulations is a plausible way to reduce simulator calls, and using the learned likelihood to generate synthetic multi-site observations is conceptually consistent with that factorization. The paper evaluates calibration and accuracy across several tasks via C2ST, and the inclusion of realistic application models is a positive sign for external validity. The introduction of a benchmark for hierarchical SBI is a nice contribution.

A central technical concern is whether improvements are due to the proposed factorization/training recipe or to architectural choices (i.e., the tokenized flow matching model) and training details. The current evidence, as described, may not isolate these factors.


Presentation:
Clear high-level motivation (hierarchical SBI, simulation cost) and an intuitively appealing decomposition (per-site likelihood, assemble multi-site).
Minor comment: Notation and readability appear to suffer early due to heavily indexed parameters; using distinct symbols for global vs local parameters may help the reader.

Significance:
Hierarchical SBI is important and widely relevant. If the approach reliably reduces simulator calls by training from single-site simulations, that can materially lower cost in regimes where simulation dominates.

Originality:
The work presents a coherent combination of (i) exploiting likelihood factorization in hierarchical SBI to decouple training from multi-site simulation, (ii) using a learned per-site likelihood as a generative surrogate to assemble multi-site observations, and (iii) a tokenized flow-matching posterior estimator intended to handle function-valued observations and variable site counts.
While the components are individually known (flow-matching posterior estimator is very similar to SinFormer), the novelty is mainly their combination. Further assessment of the individual parts (see questions) would strengthen the originality claim.

---

> ### Author Rebuttal · Authors · 2026-03-28
>
> We thank the reviewer for their assessment and for rating the work's significance as excellent. We address each question below.
>
> > What exactly is the NPE trained on in the hierarchical setting? Only the local variables?
>
> NPE is trained on a concatenation of the global and all local variables into a single vector. For example, if $\theta_g \in \mathbb{R}^2$, $\theta_l \in \mathbb{R}^3$, $n_s = 10$, then NPE is trained on $\theta \in \mathbb{R}^{32}$. Essentially, it treats the hierarchical structure as flat.
>
> > Budget accounting: when reporting "simulation budget," do you count only calls to the true simulator, or also samples drawn from the learned neural likelihood? How many neural-likelihood samples are used to train the hierarchical posterior estimator, and how sensitive are results to that number? Does the posterior estimator scale beyond 100 sites?
>
> Only true simulator calls are counted. Neural likelihood samples are orders of magnitude cheaper (6.4ms vs 11.9s for haemodynamics, Table 2), so this accounting reflects realistic decision-making where simulator cost dominates.
>
> Each training sample assembles exactly $n_s$ synthetic observations (one per site), as described in Algorithm 1 - this is determined by the hierarchical model, not a tuneable hyperparameter. If you mean to ask how performance varies with $n_s$, this is reported in the site-scaling experiment in Table 3 (bottom row of Figure 2). LF's $\ell$-C2ST degrades gracefully from $n_s = 1$ to 100 and outperforms NPE and PF on the majority of tasks at $n_s \geq 50$.
>
> Beyond 100 sites, the transformer's quadratic complexity becomes prohibitive. The MLP ablation (see below) shows comparable performance on most tasks, making it a viable drop-in for very large models.
>
> > Is the improvement over PF coming from the architecture of the estimator or the training scheme?
>
> The improved sample efficiency of our proposed method comes from Likelihood Factorisation.
>
> We have included an ablation study to demonstrate this ([tabulated results](https://anonymous.4open.science/r/tfmpe_rebuttal-9BE5/ablation_table.pdf), [ablation figure](https://anonymous.4open.science/api/repo/tfmpe_rebuttal-9BE5/file/ablation.png), [experimental setup](https://anonymous.4open.science/r/tfmpe_rebuttal-9BE5/experimental_setup_standalone.pdf)). "MLP" (LF scheme, transformer replaced) is comparable on most tasks, confirming the architecture is secondary to the factorisation strategy. The architecture facilitates functional observations (section 3.2), not the efficiency improvement.
>
> > Why is this then a good benchmark?
>
> The reviewer raises an important point for benchmarking hierarchical models. At low $n_s$, MCMC ground truth is tractable for some tasks, and we agree that including reference posteriors where feasible would strengthen the benchmark. However, a benchmark that requires MCMC ground truth for all tasks would exclude the regime where hierarchical SBI is most needed. Even at 2 sites with ground-truth initialisation, NUTS produced $\hat{R} = 1.46$ and ESS = 9 for $\beta_0$ on the SEIR model (see section A.5). It failed to converge despite extensive tuning (a gridded search of NUTS hyperparameters including tree depth and dense mass matrix adaptation, and hierarchical reparametersation to ameliorate funnel geometries). This level of per-task, model and sampler-specific intervention is not scalable to a benchmark and biases the task set towards problems where such reparameterisations happen to work. Metrics such as $\ell$-C2ST, TARP and SBC allow principled evaluation without reference posteriors, making it possible to benchmark at scales where hierarchical methods are actually needed.
>
> > Regarding example in section 3.2 and 3.3: is this real data here, if not is it possible to include some real data experiments?
>
> Sections 3.2 and 3.3 use simulated data from validated scientific simulators (a seasonal SEIR model and a 1D haemodynamics CFD solver calibrated to real arterial geometries). We prioritised simulated data to isolate the inference method from domain-specific preprocessing, but agree that real-data validation is an important next step.
>
> We will cite Figure 4 in section 3.2 and add $\text{MMD}^2 = 0.060$ ($n_\text{samples} = 400$, $n_\text{permutations} = 200$, $p < 0.005$) between TFMPE and NUTS as a quantitative complement, consistent with the close visual agreement. We have also produced haemodynamics posteriors at $n_\text{patients} = 16$ (see our response to Reviewer ogjU).

---

> > ### Author Rebuttal · Reviewer_3tvg · 2026-04-01
> >
> > I thank the authors for the their responses to the questions. The added benchmark adresses my core concern.

---

### Official Review · Reviewer_j8D1 · 2026-03-11

**Soundness:** 3
**Presentation:** 2
**Significance:** 3
**Originality:** 2
**Overall Recommendation:** 3
**Confidence:** 4

**Summary:**

This paper proposes a method for hierarchical simulation-based inference (SBI). The main idea is to reduce the number of expensive simulator calls required during training by exploiting a likelihood factorisation across sites. Instead of simulating multi-site observations directly, the method first learns a per-site likelihood model from single-site simulations. This learned likelihood is then employed to generate synthetic multi-site observations, which are used to train a posterior estimator. The posterior estimator itself is implemented using a tokenised architecture with flow matching, referred to as Tokenised Flow Matching Posterior Estimation (TFMPE). The approach is evaluated on a hierarchical SBI benchmark and two case studies (an SEIR epidemiological model and a haemodynamics simulator).

**Compliance With Llm Reviewing Policy:**

Affirmed.

**Final Justification:**

The rebuttal adresses the concern regarding presentation but here it remains difficult to assess the outcome. A clearer and more detailed discussion of the results would be appreciated. Based on the initial submission and rebuttal points, we do not see ourselves in a position to revise the score.

**Key Questions For Authors:**

Is tokenised flow matching essential for the likelihood factorisation approach, or could LF be combined with other posterior estimators?

Can the authors clarify the role of $n_{\max}$ in Figure 1 and whether it is specific to PF-NPSE or intended as a more general concept?

Could the authors provide a more detailed analysis of the SEIR experiment (Section 3.2), for example with quantitative comparisons to NUTS?

Are there plans to evaluate the haemodynamics example on a larger number of patients or to compare it with other SBI methods?

**Limitations:**

Yes

**Strengths And Weaknesses:**

The paper addresses an interesting and practically relevant problem in simulation-based inference, namely the efficient handling of hierarchical models when simulator calls are expensive. The proposed likelihood factorisation strategy is intuitively motivated and aims to reduce simulator usage by learning a per-site likelihood model that can later be used to generate synthetic observations. This idea is potentially useful in scientific machine learning settings where simulation cost dominates training. The experimental section includes both benchmark experiments and application-oriented case studies, which helps illustrate the intended use cases of the method. However, as the work mainly combines several existing ideas (likelihood modelling, flow matching, and tokenised architectures), the novelty of the paper appears moderate.

Overall, individual text passages are generally readable, but the overall structure makes it difficult to follow. Formulations require substantial background knowledge in simulation-based inference, which limits accessibility for readers outside this area. More importantly, the organisation of the paper is problematic: key ideas are introduced multiple times in different sections, while some important concepts and parameters appear without clear introduction. As a result, it is often unclear how different parts of the paper relate to each other, and I had difficulties to understand the intended connections. Moreover, central important equations are embedded in the text rather than highlighted as displayed equations, which decreases readability.

The following issues also relate to clarity and presentation. The discussion of related approaches exploiting posterior factorisation is difficult to connect to the conceptual diagram shown in Figure 1. In particular, the reviewed approaches (compositional inference, summary networks, and factorised score estimation) appear to correspond to the posterior-factorisation pipeline in that figure, but this relationship is not made explicit. Similarly, the posterior factorisation identity is introduced in the paragraph on compositional inference even though it conceptually underlies all three reviewed approaches. This identity should be introduced more generally before discussing the individual methods. Also the notation around $n_{\max}$ in Figure 1 is confusing, since the quantity appears in the figure although it is only defined in the text in the context of partially factorised score estimation.

I found it hard to understand the relationship between the three stated contributions (in the introduction). Is tokenised flow matching essential to the likelihood factorisation approach or primarily an implementation choice for the posterior estimator?

The method section provides a more formal description of the approach but partly reiterates ideas already presented earlier in the introduction, which makes the presentation feel somehow repetitive. The connection between the overview in the introduction and the formal description should be articulated more clearly.
Here, Figure 3 is not explicitly introduced in the surrounding discussion, and the role of the vector field $v_t$ produced by the architecture is not clearly explained.

In the experimental section I missed a clear discussion of the results. When describing the experimental setup, the paper introduces the configurations for LF, NPE, and PF, but it is not clearly stated that the following paragraphs contain the implementation details for the three methods being compared. The likelihood factorisation strategy itself is also described again in this section, even though it was already introduced and formalised earlier. I would suggest to explicitly reference to the earlier sections in order to avoid the impression of repetition.

The presentation of results also needs to be improved. In the SEIR case study, the main results are shown in Figure 4, but the figure is not explicitly cited in the surrounding text and the key observations are not discussed. As a result, the reader must infer the conclusions from the plot. While the experiment provides a useful sanity check by comparing the inferred posterior with NUTS, the comparison is limited to a two-site example and the discussion of the results is brief.

The haemodynamics example provides an interesting real-world case study and illustrates the potential computational advantages of the approach when simulators are expensive. However, the experiment involves only two patients, which is extremely small for a hierarchical model. While this may be acceptable as a proof of concept, it limits the strength of the conclusions. Moreover, the experiment does not include comparisons with the other inference methods evaluated earlier in the paper, which makes the section read more as an illustrative example than a systematic evaluation.

Several presentation issues are also related to figures and tables. Some figures are not explicitly cited in the text, and a few appear far from the pages on which they are discussed. In addition, the text in several figures is quite small and difficult to read. Figure 2 appears to contain results from the benchmark experiments but is placed earlier in the paper, and its axes and labels (e.g.\ the relationship between $n_s$ and the simulation budget) are not immediately clear. Figure 4 and Figure 6 also contain very small text.

As for the tables, which contain many values, I missed a description of the main trends or a summary of the results. Tables 4 and 5 are not even explicitly cited in the main text.

Finally, a few issues arise in the appendix. Appendix A.2 contains no explanatory text, which makes it unclear how it relates to the main paper. It would help to indicate which section of the paper the appendix corresponds to and which figures or tables belong to it. In Appendix A.3 the parameters $R_{T,s,o}$ and $C_{T,s,o}$ are introduced without explicitly explaining their meaning.

---

> ### Author Rebuttal · Authors · 2026-03-28
>
> We address the reviewer's point on originality, then acknowledge their concerns on presentation and then respond to their questions.
>
> The reviewer notes that the work "mainly combines several existing ideas." The novelty in our work lies in likelihood factorisation for hierarchical SBI, which we propose and validate for the first time, and the new benchmark for hierarchical SBI which facilitates progress in the field. We agree that the individual architectural components (flow matching, tokenisation) are known. However, this is the first time they have been applied in a hierarchical setting.
>
> We take on board the reviewer's concern that key ideas are introduced multiple times and that the contribution boundary is unclear. We propose:
>
>   * moving LF from section 1 to section 2 so the methodology section contains all novel contributions,
>   * making the hierarchical posterior and factorised posterior identities numbered equations,
>   * expanding section 2.1 with explicit embedding formulations and an appendix example for functional inputs,
>   * increasing text size in Figures 4 and 6,
>   * citing Figure 4 in section 3.2
>   * Tables 4–5 are cited in the caption of Figure 2, which is in the main text. We will add an additional inline citation for clarity.
>   * adding explanatory text to Appendix A.2.
>
> > Is tokenised flow matching essential for the likelihood factorisation approach, or could LF be combined with other posterior estimators?
>
> They are separable. We have included an ablation study to confirm this. Replacing the transformer with an MLP preserves performance on most tasks. Tokenisation is essential only for functional outputs (irregular time series, spatial fields). Conventional normalising flows (MAFs, NSFs) could replace the continuous flows, though they impose fixed-dimensional input constraints that make functional observations harder to handle without a separate summarisation step.
>
> The ablation study ([tabulated results](https://anonymous.4open.science/r/tfmpe_rebuttal-9BE5/ablation_table.pdf), [ablation figure](https://anonymous.4open.science/api/repo/tfmpe_rebuttal-9BE5/file/ablation.png), [experimental setup](https://anonymous.4open.science/r/tfmpe_rebuttal-9BE5/experimental_setup_standalone.pdf)) demonstrates this.
>
> > Can the authors clarify the role of $n_\max$ in Figure 1 and whether it is specific to PF-NPSE or intended as a more general concept?
>
> $n_\max$ applies to all posterior factorisation methods in the left panel of Figure 1, not only PF-NPSE. These methods sample the number of sites uniformly up to $n_\max$, requiring on average $n_\max / 2$ simulator calls per training sample. LF (right panel) always requires exactly 1. We will clarify this in the revised figure caption.
>
> > Could the authors provide a more detailed analysis of the SEIR experiment (Section 3.2), for example with quantitative comparisons to NUTS?
>
> For the 2-site case we have computed $\text{MMD}^2 = 0.060$ ($n_\text{samples} = 400$, $n_\text{permutations} = 200$, $p < 0.005$) between TFMPE and NUTS posterior samples, consistent with the close visual agreement in Figure 4. We will report this in the revised section 3.2.
>
> > Are there plans to evaluate the haemodynamics example on a larger number of patients or to compare it with other SBI methods?
>
> The haemodynamics example is a proof of concept for LF's computational savings (1850$\times$ speedup, Table 2); the benchmark suite provides the systematic cross-method comparison. To confirm calibration at larger scale, we have produced posteriors for $n_\text{patients} = 16$ ([posterior predictive](https://anonymous.4open.science/api/repo/tfmpe_rebuttal-9BE5/file/tfmpe_hemo_results_16p/posterior_predictive.png?v=40b72fb7), [global parameters](https://anonymous.4open.science/api/repo/tfmpe_rebuttal-9BE5/file/tfmpe_hemo_results_16p/posterior_global.png?v=59efbeca), [local parameters](https://anonymous.4open.science/api/repo/tfmpe_rebuttal-9BE5/file/tfmpe_hemo_results_16p/posterior_local.png?v=2f8f99ed)).
>
> However, to save computation we rely on the hierarchical SBI benchmark to report posterior quality at scale.

---

> > ### Author Rebuttal · Reviewer_j8D1 · 2026-04-02
> >
> > We acknowledge the authors rebuttal, but do not think this changes our initial assessment of the article. The suggested changes are appreciated but we are still missing a clear discussion of the results. We would need to re-review the revised manuscript before making a decision. Based on the initial submission and rebuttal points, we do not see ourselves in a position to revise the score.

---

> > > ### Author Response · Authors · 2026-04-02
> > >
> > > We thank the reviewer for their acknowledgement.
> > >
> > > To clarify, the current submission already discusses the main experimental conclusions in the main text. In Section 3.1, we discuss the LF/PF trade-off as a function of observation complexity, with Gaussian Mixture and Two Moons as contrasting cases. In Section 3.2, we discuss both the 2-site SEIR comparison to NUTS and the 100-site scalability result supported by TARP calibration.
> > >
> > > If the remaining concern is that this discussion is not presented clearly enough in the original manuscript, we accept that point and have committed to specific structural revisions. We would welcome any specifics to ensure the revision fully addresses the reviewer's expectations.

---

### Official Review · Reviewer_ogjU · 2026-03-12

**Soundness:** 3
**Presentation:** 3
**Significance:** 3
**Originality:** 3
**Overall Recommendation:** 5
**Confidence:** 4

**Summary:**

The paper proposes a tokenised flow-matching approach (TFMPE) to address the computational bottleneck of conducting simulation-based inference (SBI) in hierarchical models. Such models consist of global parameters (that control common dynamics across settings), and local parameters (that condition dynamics to site-specific observations), potentially leading to a massive parameter space - computationally expensive to simulation and perform parameter inference in. Existing amortised hierarchical SBI approaches factorise the posterior, while entailing simulations across multiple sites per training sample. The authors instead propose to learn a factorised likelihood from less computationally expensive, single-site simulations. TFMPE then leverages the learned likelihood as a surrogate simulator to generate multi-site data, which trains the final joint amortised posterior estimator. Instead of using one fixed input vector, TFMPE represents both parameters and observations as tokens, which makes it easier to work with multi-site data and function-valued observations such as irregular time series or spatial measurements.

**Compliance With Llm Reviewing Policy:**

Affirmed.

**Final Justification:**

The authors have addressed my questions during the rebuttal, and I appreciate the utility that the new method brings to the field wrt hierarchical models. Based upon the potential utility towards SBI problems involving hierarchical models, and the method being sound, and the suggested improvements during the rebuttal - my initial assessment has only been reinforced. I have taken note of other reviewers concerns - including suggestions for other baselines, but my view is that this work should be considered in the context of hierarchical models, and the authors have sufficiently benchmarked the proposed approach herein. I thank the AC, SAC and PCs for coordinating the review process well!

**Key Questions For Authors:**

Questions/Comments:
1. Do the authors have any suggestions on best-practices to maximise likelihood estimation accuracy when training the per-site local models? Also, how can a user prevent the downstream posterior estimator from over-fitting to the artifacts of the LF surrogate?
2. Arruda et. al. (2025) (cited in the paper) handle large-scale hierarchical models (250,000 groups, 750,000 parameters). TFMPE is shown to scale to 100 sites - is it possible to scale to very large number, e.g., several thousands? A discussion on scalability would be great. Apologies if I missed this somewhere.
3. I do think ablation of different components within TFMPE would further strengthen the paper.
4. You have explained the trade-offs very well in the paper, particularly in Section 4 (Discussion) on when posterior/likelihood factorisations are preferable to use. For some cases where performance drops, a suggestion would be to perhaps show estimated posteriors plotted against the ground truth. E.g., for Two Moons, it would be interest to see if the challenge is in capturing the bimodality, the crescent geometry, or both. This is not a must-do, just a suggestion.

**Limitations:**

Yes

**Strengths And Weaknesses:**

**Soundness**:
- Strengths
	- The likelihood factorisation approach is fundamentally sound and principled.
	- Treating parameters and observations symmetrically as tokens processed by a transformer is pragmatic and elegant, allowing a single architecture to handle variable data (variable-sized sets of local parameters, observations with missing values etc.)
	- Flow matching brings improved stability compared to score-based generative models (used in related works) that may entail mode-seeking behaviour
	- The validation is strong - the authors propose a hierarchical SBI benchmark suite, and put the amortised estimator against NUTS.
- Weaknesses
	- The authors point this out themselves - the performance of the amortised posterior estimator depends on the goodness of the per-site likelihood models. If the learned likelihoods are inaccurate or suffer from mode collapse, it may carry over to the amortised posterior estimator and reduce posterior quality.
	- This is relatively minor in my opinion as I appreciate the gains the proposed approach brings. However, while simulations are amortised, evaluating the PFMPE model still involves solving an ODE during inference. Since transformers entail a quadratic complexity wrt token/sequence length, generating data for a large-scale hierarchical model with hundreds of sites may still introduce substantial inference-time latency/memory bottlenecks.
	- The paper introduces 3 major components - tokenisation, flow matching and bottom-up sampling. It could be helpful to show ablations wrt these components for some benchmarks to demonstrate the relative gains owed to the different components.

**Presentation**: The paper is very well-written and presented overall.

**Significance**: I would rate the work as highly significant as it aims to address significant computational bottlenecks in hierarchical SBI, based on an elegant likelihood factorisation approach. SBI practitioners and researchers will appreciate this line of research, as well as the initiative to create a benchmark test suite.

**Originality**: The likelihood factorisation approach for hierarchical SBI is original, along with the proposed test problem benchmark suite.

---

> ### Author Rebuttal · Authors · 2026-03-28
>
> We thank the reviewer for their insightful comments. They highlight the significance and originality of likelihood factorisation and the hierarchical benchmark.
>
> > I do think ablation of different components within TFMPE would further strengthen the paper.
>
> We agree and have added an ablation study ([tabulated results](https://anonymous.4open.science/r/tfmpe_rebuttal-9BE5/ablation_table.pdf), [experimental setup](https://anonymous.4open.science/r/tfmpe_rebuttal-9BE5/experimental_setup_standalone.pdf)). We explored four variants:
>
>   * **Direct** (true simulator outputs instead of learned likelihood): $\ell$-C2ST closely tracks TFMPE across most tasks, however occasional discrepancies point to likelihood approximation error.
>   * **Joint** (unified likelihood + posterior estimator): delayed convergence on Gaussian Linear, unstable on SLCP separate estimators are preferable.
>   * **MLP** (transformer replaced with MLP): comparable on most tasks, but less stable on Gaussian Linear Uniform and SLCP. A viable alternative when functional observations are not needed, but requires per-task tuning.
>   * **Linear** (softmax attention is replaced with linear attention): performance degrades dramatically, not a promising avenue for scaling TFMPE.
>
> Results are shown in the [ablation figure](https://anonymous.4open.science/api/repo/tfmpe_rebuttal-9BE5/file/ablation.png). We will publish this to further clarify any ambiguities with respect to the separation of LF sampling from the architecture, likelihood approximation error and scaling of the transformer.
>
> > Do the authors have any suggestions on best-practices to maximise likelihood estimation accuracy when training the per-site local models? Also, how can a user prevent the downstream posterior estimator from over-fitting to the artifacts of the LF surrogate?
>
> The LF formulation is compatible with previously published likelihood based SBI techniques for mitigating this: sequential likelihood estimation (SNLE) can refine the surrogate across rounds; likelihood-ratio estimation (NRE) is equally factorisable and avoids the architectural constraints of explicit density estimation; neural variational inference (NVI) could use density evaluations rather than samples, potentially reducing variance. We have extended the discussion section to cover these directions.
>
> We recommend monitoring the per-site likelihood estimator's loss and comparing the estimated distribution against real simulations as a diagnostic for LF. If single-site calibration is poor, increasing the single-site simulation budget is far cheaper than multi-site simulation and directly improves downstream posterior quality.
>
> > Arruda et. al. (2025) handle large-scale hierarchical models (250,000 groups, 750,000 parameters). TFMPE is shown to scale to 100 sites - is it possible to scale to very large number, e.g., several thousands?
>
> Given our resources, the memory consumption of the transformer's quadratic attention in non-autoregressive decoding was a bottleneck. Memory scales as $\mathcal{O}(B T^2 h)$. For the SEIR model at $n_s = 100$ with $n_\text{obs} = 10$, the model processes $T = 1304$ tokens and the attention matrix requires $\sim 1.4\text{ GB}$ per step — tractable on a single GPU. At $n_s = 1000$, this grows to $\sim 135\text{ GB}$. However, the MLP ablation shows that replacing the transformer preserves performance on most tasks. For those interested in hierarchical models with $n_s \gg 100$, we would recommend an MLP-based vector field, as Arruda et al. adopt. We have added a discussion on scalability to the revised paper.
>
> > For some cases where performance drops, a suggestion would be to perhaps show estimated posteriors plotted against the ground truth.
>
> We agree this would be valuable. We will include posterior pair plots in the revised appendix.

---

> > ### Author Rebuttal · Reviewer_ogjU · 2026-04-01
> >
> > I thank the authors for the their responses to the questions. The suggested changes strengthen the paper further, and resolve the points I had noted.

---

### Official Review · Reviewer_26WZ · 2026-03-13

**Soundness:** 1
**Presentation:** 2
**Significance:** 3
**Originality:** 2
**Overall Recommendation:** 2
**Confidence:** 4

**Summary:**

The paper introduces TFMPE, a simulation-based inference method that leverages tokenized flow matching to learn a factorized likelihood model. The proposed method is applied in hierarchical settings where global and local parameters are meaningfully connected, and leveraging this relationship can improve simulation sample efficiency. TFMPE is demonstrated on a hierarchical SBI benchmark (adapted from a set of common non-hierarchical SBI tasks) and on real-world dynamics models, learning well-calibrated posteriors while providing unique performance-to-sample-efficiency tradeoffs compared to other baseline methods.

**Compliance With Llm Reviewing Policy:**

Affirmed.

**Final Justification:**

While the additional empirical results provided by the authors' follow-up go some way toward addressing my initial concerns, the differences in the newly provided figures make it difficult to draw reliable comparisons to those originally reported. Additionally, follow-up results appear to show somewhat non-standard behavior across methods (e.g., NPE almost invariably outperforming SNPE, when the latter often outperforms the former due to its sequentially updating proposals), calling into question the evaluation of baselines. Likelihood-based methods, such as NLE/SNLE/etc, are not included as baselines, despite the theoretical alignment (arguably more directly aligned than NPE/SNPE). Highly relevant hierarchical SBI baselines (e.g., those mentioned in the reply rebuttal comment) are not benchmarked, limiting the positioning of the work among methods most suitable for the introduced hierarchical tasks. Further, performance compared to the one hierarchical baseline, PF, is mixed (e.g., in 10 of the 18 reported task-budget combinations in Figure 2, PF reports lower $\ell$-C2ST scores than LF), leaving the performance advantage of the proposed method unclear.

**Key Questions For Authors:**

- Across benchmark settings, were other key metrics such as (non-local) C2ST, MMD, median distance, etc considered or recorded?
- For experiments with the NPE method, why were neural spline flows (NSFs) used as a density estimator? To the best of my knowledge, masked autoregressive flows (MAFs) have been the out-of-the-box default density estimator in the `sbi` package for some time now (including v0.25.0).
- Were sequential SBI methods (e.g., SNPE, SNLE, TSNPE, etc) considered as viable baselines, provided the emphasis on sample efficiency?
- In Section 2.2, it is noted that likelihood and posterior estimators were trained separately but can be trained jointly. Are there particular benefits to doing one or the other? Why was the independent training scheme decided upon?

**Limitations:**

Yes

**Strengths And Weaknesses:**

**Strengths**
- The paper's stated contributions are clear and attempt to address the difficult problem of sample efficiency in the SBI domain. Hierarchical and multi-fidelity models for SBI are a relatively underexplored frontier, and present a promising direction for making likelihood-free inference more scalable to challenging real-world settings.
- The factorized likelihood approach appears sound and serves as a viable alternative to direct posterior factorization for hierarchical settings.
- The approach is topical and builds on recent advances in SBI (e.g., Simformer) that increase the portability of learned models and unify the modeling of multiple probabilistic forms.

**Weaknesses**
- The reported results on the core benchmark suite are missing key comparisons to several relevant SBI methods, making the evaluation of the proposed method difficult to contextualize in the broader LFI landscape. In particular, it would seem important to include comparisons to methods with heavy architectural overlap, namely:

  * Flow-matching or score estimation methods like FMPE[1] and NPSE[2], and
  * Recent tokenized approaches like Simformer[3], from which the model appears to borrow inspiration for its tokenization scheme

  Comparisons to these methods would be necessary to better judge the impact of the technical contributions of TFMPE, serving as key ablations for the approach's architectural components. Additionally, when sample efficiency is a primary concern, sequential methods (e.g., SNPE, SNLE, and common variants) often outperform non-sequential NPE by allowing proposal updates across several rounds of inference. Comparisons to these methods would be critical to analyze the impact of the proposed approach when sample efficiency is a key factor.
- The presentation of the paper is somewhat difficult to follow. Much of the pertinent method details are found in the introduction, while the methodology section is fairly light on technical detail, making it difficult to understand the boundary of the proposed approach. Additionally, the tokenisation scheme is quite vague, leaving some ambiguity behind how variable identifiers are embedded and how multi-dimensional/functional inputs are handled. Even if there is heavy overlap with pre-existing work (e.g., Simformer's tokenization scheme), it seems important to explicitly reproduce those details or indicate how the proposed scheme differs. In my opinion, Figure 3 does little to further contextualize the pipeline, providing only a high-level architecture overview that lacks emphasis of the method's novel contributions.
- Minor remarks:
  * I found the task specifics (e.g., those presented in Table 1) and full modeling details (like those in the right-hand column of page 5) hindered the overall paper flow, and may be best delegated (at least in part) to the appendix.
  * Figure 4 does not appear to be referenced in the main paper body.

[1]: Wildberger, Jonas, et al. "Flow matching for scalable simulation-based inference." Advances in Neural Information Processing Systems 36 (2023): 16837-16864.
[2]: Geffner, Tomas, George Papamakarios, and Andriy Mnih. "Compositional score modeling for simulation-based inference." International Conference on Machine Learning. PMLR, 2023.
[3]: Gloeckler, Manuel, et al. "All-in-one simulation-based inference." arXiv preprint arXiv:2404.09636 (2024).

---

> ### Author Rebuttal · Authors · 2026-03-28
>
> We thank the reviewer for suggesting new baselines. These baselines were not originally included because they do not address hierarchical structure, they estimate posteriors or scores in the flat setting. Our contribution targets the simulation bottleneck specific to hierarchical models. Nevertheless, we agree that comparing against strong non-hierarchical baselines is essential to isolate the gains from factorisation, and we have done so.
>
> We believe the new results warrant reconsideration of the soundness and originality scores. We have added FMPE (MLP and Transformer variants), Simformer (with posterior estimation - we believe this is can serve as a proxy for NPSE as well), and SNPE (5 rounds) to the benchmark ([tabulated results](https://anonymous.4open.science/r/tfmpe_rebuttal-9BE5/review_table.pdf), [benchmark figure](https://anonymous.4open.science/api/repo/tfmpe_rebuttal-9BE5/file/lc2st.png), [experimental setup](https://anonymous.4open.science/r/tfmpe_rebuttal-9BE5/experimental_setup_standalone.pdf)). Across all six tasks at $n_s = 50$, the non-hierarchical methods tend to plateau at $\ell$-C2ST $\geq 0.15$ regardless of simulation budget. Only the factorised methods (Likelihood Factorisation - LF and Posterior Factorisation - PF) consistently show improvement with increased budget.
>
> Our paper's claims:
>
> (1) that LF can improve sample efficiency for hierarchical models,
> (2) that TFMPE + LF can produce well calibrated posteriors for hierarchical models with functional observations, and
> (3) that TFMPE + LF leads to computational efficiency gains for hierarchical models
>
> are supported by the benchmark results (now with the requested baselines), the NUTS comparison and the TARP analysis for the SEIR model, and the telemetry for the haemodynamics model.
>
> LF sampling and the hierarchical benchmark are each original contributions. The tokenised architecture is not a novelty claim in its own right but is original in the hierarchical setting, where it enables functional observations (section 3.2).
>
> The reviewer notes that comparisons to FMPE and Simformer would serve as "key ablations for the approach's architectural components." Beyond the new baselines above, we have conducted an ablation study ([tabulated results](https://anonymous.4open.science/r/tfmpe_rebuttal-9BE5/ablation_table.pdf), [ablation figure](https://anonymous.4open.science/api/repo/tfmpe_rebuttal-9BE5/file/ablation.png)) isolating the architecture from the factorisation. The MLP ablation (LF sampling, transformer replaced with MLP) preserves performance on most tasks, confirming that LF sampling, not the transformer, drives the efficiency gains.
>
> We acknowledge the unclear contribution boundary. We will move LF from section 1 to section 2 so that all novel contributions sit in the methodology, expand tokenisation details in section 2.1 (with an appendix diagram for functional inputs), add architectural detail to Figure 3, and cite Figure 4 in section 3.2. We will tighten Table 1 but retain it as the benchmark is a stated contribution.
>
> > Across benchmark settings, were other key metrics such as (non-local) C2ST, MMD, median distance, etc considered or recorded?
>
> Yes — we justify $\ell$-C2ST on page 5, line 270. Standard C2ST and MMD require ground-truth posteriors, which become prohibitively expensive at hierarchical scales. Median distance is a point-estimate metric that can mislead unless the distance distribution under the true posterior is known.
>
> > For experiments with the NPE method, why were neural spline flows (NSFs) used as a density estimator? To the best of my knowledge, masked autoregressive flows (MAFs) have been the out-of-the-box default density estimator in the sbi package for some time now (including v0.25.0).
>
> Our benchmark was adapted from Lueckmann et al., who used NSFs. The new baselines now span flow matching, tokenised normalising flows, and sequential NSFs. No non-hierarchical estimator matches the factorised methods regardless of architecture.
>
> > Were sequential SBI methods (e.g., SNPE, SNLE, TSNPE, etc) considered as viable baselines, provided the emphasis on sample efficiency?
>
> Yes. SNPE (5 rounds) is included in the new results. At $n_s = 50$ with small budgets, the per-round budget is insufficient to train. Where SNPE can run, it plateaus with the other non-hierarchical methods.
>
> > In Section 2.2, it is noted that likelihood and posterior estimators were trained separately but can be trained jointly. Are there particular benefits to doing one or the other? Why was the independent training scheme decided upon?
>
> The "Joint" ablation (see figure above) shows that the joint objective leads to delayed convergence on Gaussian Linear tasks and instability on SLCP. It also suffered OOM on SIR. Separate estimators benefit from simpler optimisation and we recommend them as the default.

---

> > ### Author Rebuttal · Reviewer_26WZ · 2026-04-04
> >
> > I appreciate the authors' detailed response and additional empirical evaluations. The extensive baseline reports provide meaningful context comparing the proposed approach to common baselines, and address the top level concern regarding the scope of evaluation. However, there appear to be significant differences in reported performance in the provided plots [here](https://anonymous.4open.science/api/repo/tfmpe_rebuttal-9BE5/file/lc2st.png) compared to those originally presented in Figure 2 of the paper. The experimental setup (e.g., the three reported simulation budgets) appears to be identical, but across each task, the $\ell$-C2ST scores are often very different at each budget. For instance, in Hierarchical SLCP of the newly attached figure, **PF** is very close to `0.0` for all sample sizes, whereas in Figure 2 it never achieves a score lower than `0.05`. This is one of many discrepancies between the two figures; can the authors explain why there appears to be such a large difference here?
> >
> > The soundness evaluation is primarily a result of a limited presentation of the methodology and a lack of relevant proofs under the proposed formulations commonly observed for SBI methods (e.g., proof of convergence under the proposed sampling scheme). I appreciate this has been acknowledged in the rebuttal, but there is still outstanding uncertainty regarding the general formulation, scope, and theoretical guarantees of the proposed methodology.

---

> > > ### Author Response · Authors · 2026-04-04
> > >
> > > We thank the reviewer for their notes and acknowledging our rebuttal.
> > >
> > > > The experimental setup (e.g., the three reported simulation budgets) appears to be identical ...
> > > > can the authors explain why there appears to be such a large difference here?
> > >
> > > The experimental setups differ in the $\ell$-c2st classifier protocol. The rebuttal figures use a 10-fold cross-validated ensemble classifier (detailed in Section A.1 of the experimental setup we linked in the rebuttal), while the original submission used a single-fold, single-hidden-layer classifier (detailed in Section A.6). The cross-validated protocol reduces variance, most visibly for SLCP where the original error bars were wide. We should have flagged this change more prominently in the rebuttal and regret the confusion.
> > >
> > > Our paper's claims concern the relative performance between methods, which remain largely unchanged. LF and PF consistently outperform non-hierarchical baselines across tasks and budgets and support our discussion on their trade-offs in the main text (concerning likelihood and posterior complexity).
> > >
> > > We are adopting the stronger cross-validated protocol throughout our revised manuscript at the time of writing.
> > >
> > > > a lack of relevant proofs under the proposed formulations commonly observed for SBI methods (e.g., proof of convergence under the proposed sampling scheme)
> > >
> > > We appreciate the reviewer raising theoretical guarantees. The only approximation introduced by LF is the (factorised) neural likelihood estimator, the same approximation present in NLE (Papamakarios et al., 2019), for which convergence is an open problem. To our knowledge, no existing hierarchical SBI method, including Heinrich et al. (2023), Arruda et al. (2025), Radev et al. (2023), and Habermann et al. (2024), provides convergence guarantees for its factorisation or sampling scheme. We acknowledge this as a limitation of the field at large and an important direction for future work.
> > >
> > > We have addressed the reviewer's concerns: (a) the original soundness concern regarding missing baselines (FMPE, Simformer, SNPE now included), (b) committed to specific structural revisions to improve the presentation of the methodology, (c) the figure discrepancy - highlighted the difference in our protocols, (d) contextualised theoretical guarantees in the field. We believe these additions and clarifications substantiate the paper's claims.

---

### Decision · Program_Chairs · 2026-04-30

**Decision:**

Accept (regular)

**Comment:**

The average rating is 3.75 with split scores (5, 5, 3, 2), placing this paper clearly at the borderline with reviewers in each camp. The paper proposes likelihood factorisation (LF) for hierarchical SBI, a tokenised flow matching estimator (TFMPE), and a benchmark suite.

Reviewers ogjU and 3tvg find the contribution sound and significant for the SBI community; 3tvg confirmed post-rebuttal that "the added benchmark adresses my core concern." On the other side, j8D1 maintains weak reject over presentation issues (repetition, unclear contribution boundary, uncited figures) and states they "would need to re-review the revised manuscript." Reviewer 26WZ is the sharpest critic on evaluation grounds: missing NLE/SNLE baselines and the observation that "PF reports lower C2ST scores than LF" in over half the task-budget pairs. 3tvg also noted "due to the many changes, another round of reviews might be appropriate."

The rebuttal is substantial. The authors added FMPE, Simformer, and SNPE (5 rounds) as baselines, and ran an ablation showing that an MLP variant preserves performance, isolating the gains to LF rather than the transformer. This directly resolves 3tvg's central concern. 26WZ's concerns are partially valid (NLE is a genuine gap given its theoretical alignment with LF) but partially overstated: the paper does not claim LF dominance over PF; Section 3.1 explicitly discusses the LF/PF trade-off as complementary methods. The core claim, that factorised methods outperform non-hierarchical baselines on hierarchical tasks, is supported by the added baselines.

Weighing the evidence, I lean towards acceptance. The contribution, LF plus a dedicated benchmark, is original and addresses a real bottleneck for a community where simulation cost dominates. The rebuttal was responsive and the ablation isolates the methodological contribution from the architectural one. The remaining gaps (missing NLE baseline, promised presentation revisions) are additive rather than fundamental; they would strengthen the paper but do not invalidate its claims.